# Symmetry-adapted Markov state models of closing, opening, and desensitizing in α7 nicotinic acetylcholine receptors

Yuxuan Zhuang[1], Rebecca J. Howard ®[1] & Erik Lindahl[1,2] ✉

α7 nicotinic acetylcholine receptors (nAChRs) are homopentameric ligand-gated ion channels with critical roles in the nervous system. Recent studies have resolved and functionally annotated closed, open, and desensitized states of these receptors, providing insight into ion permeation and lipid binding. However, the process by which α7 nAChRs transition between states remains unclear. To understand gating and lipid modulation, we generated two ensembles of molecular dynamics simulations of apo α7 nAChRs, with or without cholesterol. Using symmetry-adapted Markov state modeling, we developed a five-state gating model. Free energies recapitulated functional behavior, with the closed state dominating in absence of agonist. Open-to-nonconducting transition rates corresponded to experimental open durations. Cholesterol relatively stabilized the desensitized state, and reduced open-desensitized barriers. These results establish plausible asymmetric transition pathways between states, define lipid modulation effects on the α7 nAChR conformational cycle, and provide an ensemble of structural models applicable to rational design of lipidic pharmaceuticals.

Pentameric ligand-gated ion channels (pLGICs) play a crucial role in the transmission of signals within the nervous system[1–3]. Among pLGICs, nicotinic acetylcholine receptors (nAChRs) constitute a family that can be activated by the neurotransmitter acetylcholine (ACh). These receptors are widely distributed across different species and can be found at both neuromuscular junctions and synaptic clefts[4]. The nAChRs exhibit heterogeneity through the assembly of different combinations of subtypes, among which 16 have been identified within the human genome. These receptors demonstrate diverse physiological and pharmacological properties. Notably, the α7 subtype has the ability to form homomeric receptors and is abundant in the central nervous system, predominantly located pre- or peri-synaptically. Dysfunction of the α7 subtype can contribute to various neurological and inflammatory disorders[5–9]. Consequently, this subtype has emerged as a prominent target for drug development. However, the lack of a comprehensive understanding of its gating mechanism has hindered the development of effective therapeutic interventions in clinical settings[10,11].

The gating process of nAChRs primarily encompasses three functional states: closed, open, and desensitized[12]. The resting, or closed, state predominates when no agonists are bound to the receptors. Upon binding of agonists in the extracellular domain (ECD), the receptors undergo a subsequent conformational change, leading to the opening of the hydrophobic pore in the transmembrane domain (TMD) and allowing cations to pass through. Following prolonged exposure to agonists, the pore closes again, entering a distinct non-conductive desensitized state. The α7 subtype also displays a distinct kinetic profile compared to other subtypes[13]. It undergoes desensitization significantly faster in response to ACh, even faster than some of the solution exchange rates in electrophysiology recordings[13]. This rapid desensitization poses challenges in achieving a consensus estimation of the open duration profile using various techniques.

[1]Department of Biochemistry and Biophysics, Science for Life Laboratory, Stockholm University, Solna, Stockholm, Sweden. [2]Department of Applied Physics, Swedish e-Science Research Center, KTH Royal Institute of Technology, Solna, Stockholm, Sweden. ✉e-mail: erik.lindahl@dbb.su.se

Recently, advances in cryogenic electron microscopy (cryo-EM) have enabled the resolution of all three states (closed, open, and desensitized) of the $\alpha$ 7 subtype of nAChRs[14,15]. We subsequently validated and functionally annotated these structures using molecular dynamics (MD) simulations[16]. These studies have provided detailed insights into the structural basis of gating and the critical residues that govern the ion permeation pathway of $\alpha$ 7 nAChRs. However, the precise manner in which these states are interconnected, whether symmetrically or in an asynchronous fashion, and the regulatory mechanisms underlying the dynamics of the gating process remain unclear.

Various techniques in electrophysiology have been employed since the 1970s[17] to investigate conformational changes in nAChRs[18]. For instance, the phenomenon known as "nachschlag shutting"−brief closures during a burst of openings[19,20]−can be attributed to a non-conductive state preceding channel opening. It is distinct from the closed state and is referred to as "flipped" or "primed" in the literature[12]. This flipped state has been functionally observed in several members within the pentameric ligand-gated ion channel (pLGIC) superfamily[21,22]. Interestingly, the transition between the flipped state and the open state has been found to be independent of agonist binding. In the case of glycine receptors, another member of the pLGIC superfamily, both full agonists and partial agonists exhibit similar shut time distributions originating from the flipped state[21]. A recent study on nAChRs, using kinetic modeling of an ancestral $\beta$ homopentamer, placed the flipped state at the center of the gating cycle[23]. The authors further proposed that the agonists that agonists might function in derepression−while appearing as activation−the intrinsic profile inherited from ancestral nAChRs[23]. Despite these insights, the precise configuration of the flipped state and its relationship to other functional states remain unknown, and the transient nature of states like this can make them challenging to pursue with structural biology.

Lipids have the ability to modulate the gating of nAChRs through at least two mechanisms: altering bulk lipid properties or directly binding to the protein[24–26]. For $\alpha$ 7, both these mechanisms may play a role. Structural and computational studies showed local membrane compression in the open state[14,16]. We also identified potential binding sites for cholesterol within the $\alpha$ 7 receptor TMD. Interestingly, PNU-120596 (PNU), a positive allosteric modulator (PAM), shared the same binding site as cholesterol at the protein-membrane interface. Based on these findings, we proposed a modulation mechanism for PNU in which it displaces cholesterol, thereby promoting receptor opening while preventing desensitization. These results highlight the insights provided by MD simulations in understanding the binding of lipidic modulators to various functional states and their short-range perturbation effects. However, it should be noted that the timescale of the gating process, which occurs over milliseconds or longer, is beyond the reach of conventional MD simulations[27].

Markov state models (MSMs) may provide a useful approach for studying the kinetics of biological systems[28,29]. MSMs are closely related to continuous-time kinetic modeling[30–32], but they represent the system as a discrete-time Markovian process, in which information can be described solely by the transition probability matrix, which defines the probabilities of transitioning between different states over a discrete lag time, denoted as $\tau$. The MSM framework has gained popularity as it enables the connection of multiple short MD simulations and helps bridge the timescale gap to experimental studies of kinetics[33]. In an MSM, conformational changes are considered as Markovian or memoryless transitions between discrete states. To construct an MSM, one must determine an appropriate lag time $\tau$ and the number of microstates so that the relaxation time within each microstate is shorter than $\tau$ for the model to be valid. This requirement poses challenges when modeling large proteins[34,35]. Modeling proteins as large as pLGICs often necessitates either a large number of microstates or a long lag time, which contradicts the advantages offered by the MSM framework.

In this study, we utilize MSMs to investigate the gating mechanism of $\alpha$ 7 nAChRs after seeding simulations along three approximate gating transition pathways which encompass experimentally resolved closed, open, and desensitized states. Taking advantage of the symmetrical nature of $\alpha$ 7 nAChRs, we construct MSMs using a reduced set of microstates. By analyzing the MSMs we are able to map the free energy landscape of the entire functional cycle of the $\alpha$ 7 nAChR, and in particular identify local structural transitions associated with the presumed functional gating cycle, generally extending from the ECD to the TMD through the coupling region. The simulations allow us to explore the role of symmetry in the gating process, and by repeating simulations of the same system in the presence of cholesterol we are able to quantitatively assess its modulating effect on the gating of $\alpha$ 7 nAChRs.

## Results

### Delineating degenerate gating dynamics with symmetry-adapted time-lagged independent component analysis

Recently validated cryo-electron microscopy (cryo-EM) structures of $\alpha$ 7 in closed (PDB ID 7KOO), open (PDB ID 7KOX), and desensitized (PDB ID 7KOQ) states[14] were used as initial endpoints in this study. To explore the conformational space of the protein, the Climber algorithm[36] was employed. This algorithm constructed initial pathways in all directions, resulting in a total of six pathways and a total of 150 seeds that mapped the conformational space of interest (Fig. 1A, Supplementary Fig. 1). Each seed was embedded in a lipid bilayer, equilibrated, and further simulated without any restraints or ligands for 1–2 μs (apo system).

To address the statistical error and lack-of-sampling problem associated with exploring the entire gating space, which presumably involves asymmetric conformational changes in any or all of the five subunits (Fig. 1B), we developed symmetry-adapted time-lagged independent component analysis (SymTICA) to extract degenerate conformational changes. SymTICA is an extension of TICA[37] that specifically accounts for the symmetry present in the system (see "Methods" for detailed description). Similar to TICA, SymTICA aims to capture the slowest dynamics of the system by projecting the high-dimensional conformational space onto a low-dimensional space. Specifically, it identifies low-dimensional features within each subsystem that exhibit symmetry−in this case, the $C_\alpha$ contacts within one subunit and with neighboring subunits. The resulting low-dimensional features, referred to as sub-independent components (subICs), were then combined to construct the full independent components (ICs) by performing a direct sum operation (Fig. 1C). This approach enables the extraction of collective conformational changes that respect the underlying symmetry of the system.

In our analysis, we projected all conformational snapshots obtained from our MD simulations onto the space defined by the first two ICs (IC1-IC2), which was constructed using 1610 $C_\alpha$ contacts (Supplementary Fig. 2). This allowed us to visualize the conformational landscape and identify the dominant conformational changes associated with the gating process. IC1 primarily captured the separation between the closed state and the open/desensitized states, while IC2 primarily captured the transition between the desensitized state and closed/open states (Fig. 1D, E, Supplementary Fig. 3). No major state differentiation was apparent in projections along shorter-timescale components (IC3-6) (Supplementary Fig. 3). The observation that the projections along IC1 and IC2 were distinct from subIC1-IC2 indicated an asymmetric nature of the transitions (Fig. 1D, E). If the transitions were symmetric, we would expect the two projections to be nearly identical. The transitions between all three states appeared to be well-connected in the IC1-IC2 space, suggesting a continuous pathway between different gating states. This provided evidence that we could

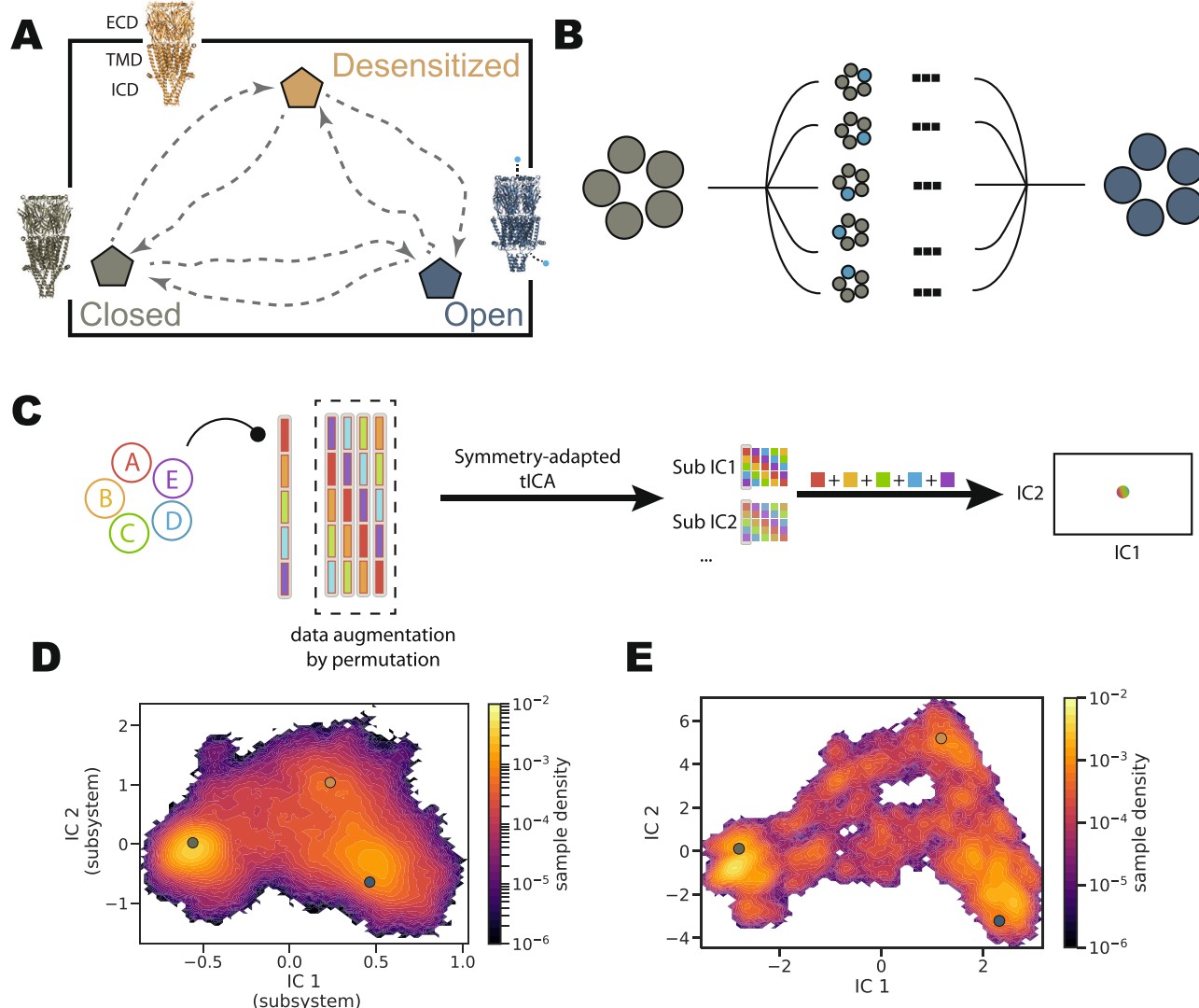

**Fig. 1 | Symmetry-adapted time-lagged independent component analysis (SymTICA) to model nAChR gating. A** Initial pathways generated using Climber[36] between three structural models: closed (PDB ID 7KOO), open (PDB ID 7KOX), and desensitized (PDB ID 7KOQ). **B** Asymmetric pathways during transitions between different states of a symmetric pentamer. These pathways exhibit the same kinetic properties. **C** Illustration of SymTICA applied to a symmetric pentamer system. Trajectory features are arranged in a block that respects the system's symmetry. Four additional pseudo-trajectories are generated by permuting the blocks. Each block is then decomposed into sub-independent components (subICs), which combine to form the full ICs through direct summation. **D** Projection of all conformational snapshots onto the subIC1-IC2 space. The three structural models are projected on the density map (gray: closed; blue: open; yellow: desensitized). **E** Projection of all conformational snapshots onto the IC1-IC2 space. The three structural models are projected on the density map (gray: closed; blue: open; yellow: desensitized).

proceed with constructing MSMs to quantitatively describe the dynamics of the entire gating process.

## Markov state model defines five conformational states in the gating cycle

Following the discretization of the full ICs space into 1000 microstates (Supplementary Fig. 7), we performed a further coarse-graining step to identify five metastable macrostates (Fig. 2A). The validity of the MSM is confirmed through the application of both the Chapman-Kolmogorov (CK) test[38] and the enumeration of transitions between states (Supplementary Fig. 4, Supplementary Fig. 5, Supplementary Fig. 6, Supplementary Fig. 8). Three of these macrostates corresponded to the well-known canonical states: closed (C), open (O), and desensitized (D). Additionally, we identified two intermediate macrostates that bridged transitions between canonical states. One intermediate macrostate, referred to here as flipped (F), connected the closed and open states, while the other intermediate macrostate (I) connected the desensitized and closed states (Fig. 2A).

Interestingly, an additional cryo-EM structure obtained in the presence of EVP-6124 and PNU, with a distinctive conformation compared to those used for initial pathway generation, fell into the basin corresponding to the F state (Fig. 2A, B, Supplementary Fig. 7). It is worth noting that this structure was not used in any of our simulations or the construction of the MSM. The structure was previously described as a pre-open state, characterized by a more constricted pore compared to the apparent open state[15]. This observation suggests that the SymTICA analysis and the resulting macrostate model were able to capture relevant conformational states, even when not used to seed the transitions.

The free-energy surface obtained from MSM analysis of the apo system contained distinct basins corresponding to different conformational states (Fig. 2B). State C mapped to the deepest basin,

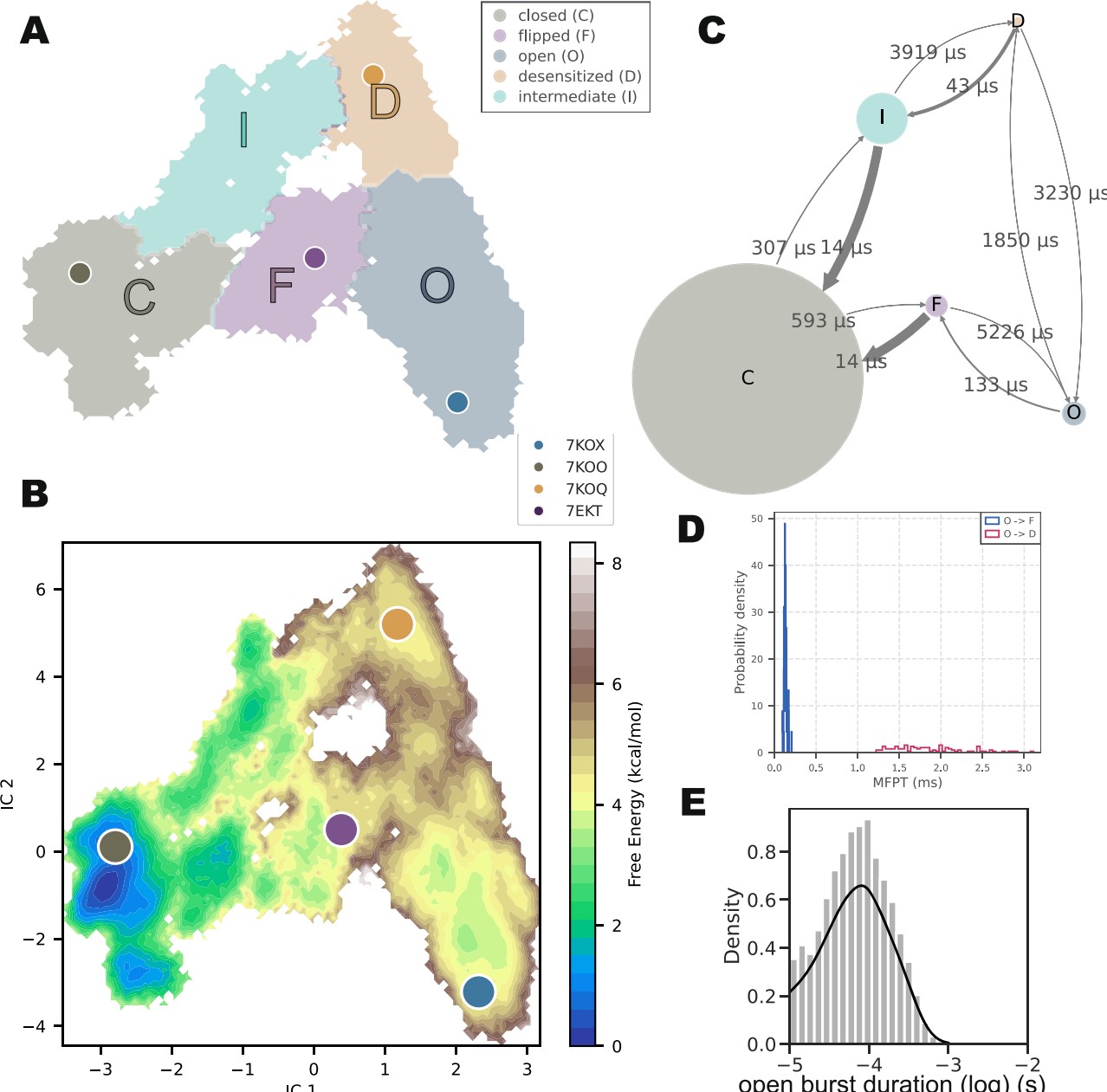

**Fig. 2 | MSM and kinetics of the apo system. A** Mapping of a coarse-grained MSM with five macrostates onto IC1-IC2 coordinates. Four structural models are projected onto the plot. **B** Free energy surface of gating projected onto IC1-IC2 coordinates. The color map represents the free energies in different regions. Four structural models were projected. **C** Mean first passage times (indicated by arrows) between consecutive macrostates, with circle size indicating the relative stability of each macrostate. For simplicity, only passages along the predominant gating cycle are shown; for quantification of all transitions, see Supplementary Fig. 8. **D** Histogram of mean first passage times between states O and F (blue) and between states O and D (orange), estimated from Bayesian MSM (100 samples). **E** Histogram of open burst durations estimated from the MSM.

while state O had a free energy of 2.8 kcal/mol higher. Perhaps surprisingly, state D was also energetically higher than state C by 3.9 kcal/mol, as detailed in the final section of the results. State F and state I were higher in free energy by 2.9 kcal/mol and 1.9 kcal/mol, respectively, compared to state C (Table 1). Thus as anticipated for the unliganded condition, the majority of channels resided in their closed state.

Kinetically, the MSM allowed us to characterize conformational transitions up to millisecond scales, which can also be illustrated as a virtual trajectory of stitched-together shorter trajectories (Supplementary Movie 1). The mean first passage times (MFPTs) between different states show how the system has a strong preference for the closed state C (Fig. 2C, Supplementary Fig. 7). The 133-μs open-to-flipped MFPT (Fig. 2C, D) was notably comparable to the experimentally measured opening duration of 100 μs in the presence of agonists[39,40]. Although opening duration time cannot be trivially obtained in the absence of agonist for $\alpha$7, it is plausible that these values indicate an inherent opening rhythm of these receptor subtypes[23,41]. Furthermore, MSM analysis allowed us to quantify the open burst duration, which represents the average length of spontaneous channel opening events (Fig. 2E). Although there is limited electrophysiological evidence for spontaneous opening in $\alpha$7−likely due to its short interval−the open burst duration obtained from MSM analysis aligned with the shortest component of the opening duration

observed in previous single-channel recordings[42]. These findings indicate that the MSM captured essential dynamics of the gating process. Consequently, the MSM was employed to investigate conformational changes in the receptor in more detail.

## Table 1 | MSM free energy differences between states in apo and CHOL system

| state | G (kcal/mol) apo | G (kcal/mol) CHOL |
|---|---|---|
| closed | 0.0 | 0.0 |
| flipped | 2.9 | 1.8 |
| open | 2.8 | 1.5 |
| desensitized | 3.9 | 1.5 |
| intermediate | 1.9 | 1.3 |

## Sequential conformational changes in gating

To comprehend the structural transitions that occur across the presumed gating cycle of the channel—from closed to open, to desensitized, and finally back to closed—as likely observed in electrophysiology experiments[12], we focused on this sequence of conformational changes and the corresponding dynamics. We first assessed the extent of pore hydration in each macrostate (Fig. 3A). In state O, the pore was fully hydrated, consistent with our previous calculations that it allows $Ca^{2+}$ ions to permeate[16]. As the channel transitioned to the desensitized state, the inner mouth of the pore became constricted. Additionally, in states I and C, the hydrophobic region at position 9′ exhibited dewetting. Notably, state F exhibited heterogeneous pore hydration—never as hydrated as state O—and should thus be non-conductive. Sojourns in such a state could result in brief closures during a burst of opening[19,20].

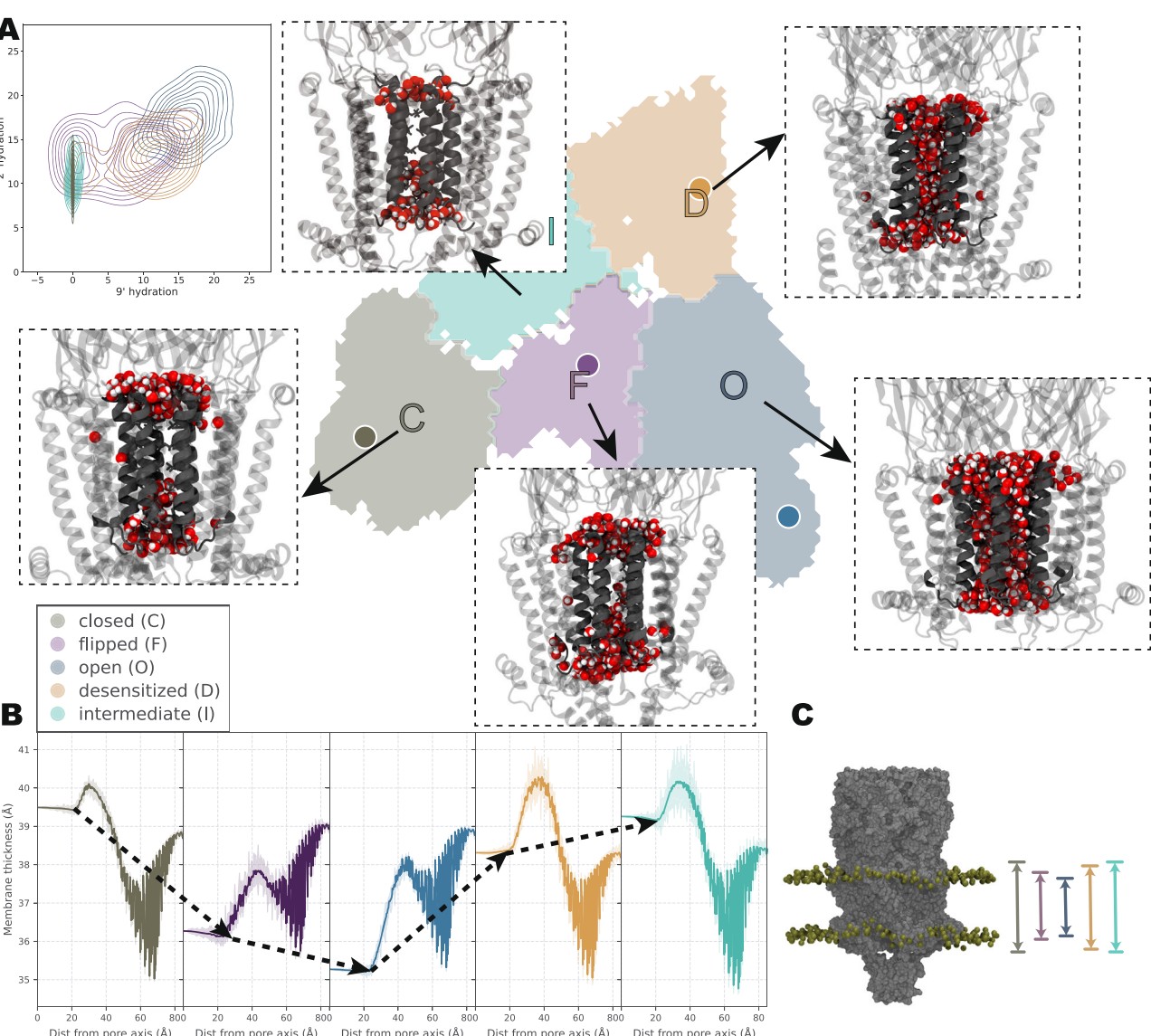

**Fig. 3 | Characteristics of macrostates: pore hydration and membrane compression. A** Snapshots of pore hydration in each macrostate, visualizing water molecules as spheres. The inset shows a histogram of the number of water molecules around the 9′ residue versus the 2′ residue. This analysis was performed within a cylindrical region centered at each residue, extending ±2 Å along the pore axis. **B** Membrane thickness profiles in each macrostate were obtained by radially averaging the membrane thickness centered at the pore axis. The shaded area indicates the 95% confidence interval. **C** Illustration of the protein embedded within the membrane, with phosphorus (P) atoms depicted as spheres. The membrane undergoes gradual compression as the protein transitions from state C to state F and then to state O. It relaxes back from state O to state D, and further transitions through state I back to state C.

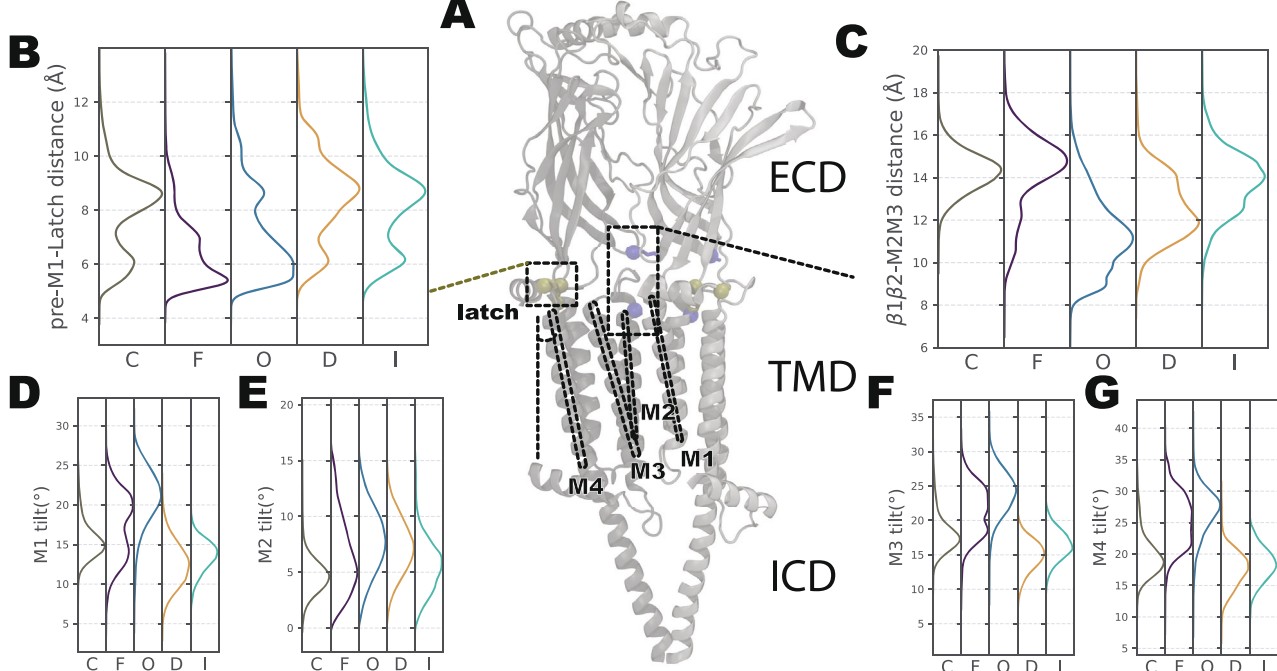

**Fig. 4 | Sequential conformational changes of coupling region and transmembrane domain. A** Visualization of two consecutive subunits, highlighting the contact between the pre-M1 loop and latch helix (shown in tan) and the contact between the $\beta 1$-$\beta 2$ loop and M2-M3 loop (shown in blue). The M1, M2, M3, and M4 helices are labeled with dashed lines. **B** Histogram showing the distribution of contacts between the pre-M1 loop and latch helix for each macrostate. **C** Histogram showing the distribution of contacts between the $\beta 1$-$\beta 2$ loop and M2-M3 loop for each macrostate. (**D**, **E**, **F**, **G**) Histograms showing the distribution of tilt angles for the M1, M2, M3, and M4 helices, respectively, in each macrostate. All distributions are weighted by the MSM weights.

Furthermore, we observed that the membrane thickness varied with the receptor state. As the system progressed from C to F the local membrane underwent compression, resulting in a subsequent decrease of ~4 Å in state O, which aligns with our previous findings[16]. The membrane thickness then gradually relaxed back from state O to state D and I, ultimately resembling the thickness observed in state C (Fig. 3B, C).

The observed differences in bulk membrane properties and pore profiles may be attributed to the conformational variance among different states. Conformational changes in the ECD were transmitted to the pore through the coupling region between the domains (Fig. 4A–C). Specifically, the distance between the pre-M1 loop and the latch region[14] at the C-terminal end displayed a similar distribution for states C, I, and D, characterized by a loosening of contact (Fig. 4B). In contrast, both states F and O exhibited a tightening of this contact, which may account for subsequent conformational changes observed in the M1, M3, and M4 helices within the TMD (Fig. 4D, F, G). The tilt of these helices demonstrated distinct differences between state C/D/I and state O/F. These conformational changes within the TMD could play a role in membrane compression and facilitating the opening of the pore through the tilting of the M2 helix (Fig. 4E).

Interestingly, the contact between the $\beta 1$-$\beta 2$ loop in the ECD and the M2-M3 loop in the TMD exhibited asynchronous behavior compared to the contact between the pre-M1 loop and the latch region (Fig. 4A, C) that happened relatively early during opening. Specifically, these two loops were not in contact except in state O and started to disengage after transitioning to state D. This conformational change was correlated with the tilting of the M2 helix, which in turn contributed to the opening of the pore.

**Symmetry in gating transitions**

The SymTICA methodology utilizes the dynamic variances of sub-independent components (subICs) to capture the symmetric

information present in the system. The analysis of symmetry-aware multidimensional scaling (MDS) (see the "Method" section for a detailed description) mapping and the standard deviation of subIC1 revealed distinct conformational characteristics of the different states. In state C, MDS mapping and subIC1 standard deviation analysis indicated a conserved symmetric conformation (Fig. 5A, Supplementary Fig. 9). The majority of conformations in states O and D also exhibited symmetry, although some heterogeneity was observed. On the other hand, state F and I predominantly adopted asymmetric conformations.

To further investigate the conformational differences, we identified the major local free-energy minimum (with a population above 10%) using InfleCS[43]. Among these minimum-energy structures representing different states, we visualized differential pore symmetry via the pentagonal belt located at each channel's hydrophobic gate (Fig. 5B). The vertices of each pentagon correspond to $C_{\alpha}$ atoms of the 9' residues. In a symmetric conformation, the geometry would be regular; in an asymmetric pore, it would be more or less distorted. In states C and D, only one near-symmetric conformation was observed, while state O exhibits a mixture of near-symmetric conformations. In contrast, states F and I predominantly displayed a heterogeneous distribution of asymmetric conformations, which was also reflected in the edge length distributions (Fig. 5C).

**Free-energy landscapes capture stabilizing effect of cholesterol**

Finally, we conducted additional computational experiments to examine the effect of cholesterol (CHOL) by re-embedding the initial conformations into a 33% CHOL membrane system (CHOL system). Five cholesterols were docked near the predicted intersubunit cholesterol binding site[16] (Fig. 6A) to ensure sufficient sampling of bound states. In the desensitized state, this binding site was previously shown to be selectively accessible to cholesterol, indicating a specific interaction between cholesterol and the receptor in this state. Conversely, in the open state, the same binding site can be occupied by the PAM

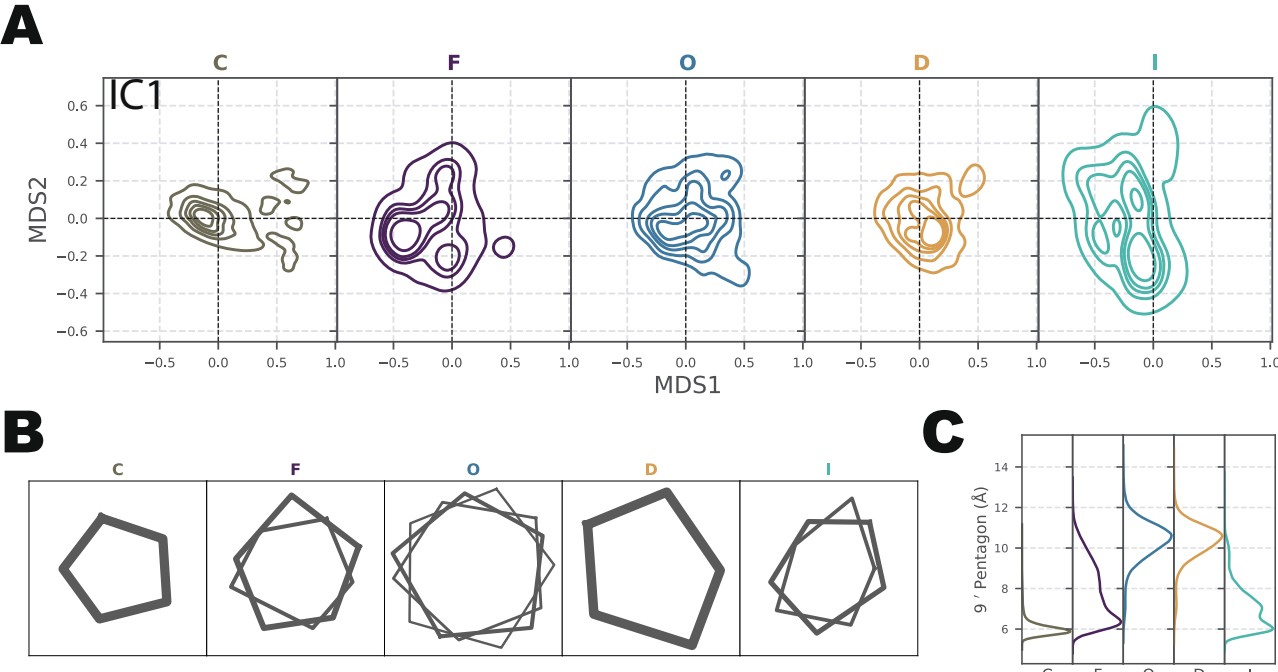

**Fig. 5 | Asymmetric intermediate states. A** Symmetry-aware multidimensional scaling (MDS) projected sub Independent Components (IC1) onto the MDS1-MDS2 space. The zero point represents perfect symmetry with mean value, while a shifted distribution suggests sampling of asymmetric conformations. For instance, in the C system, only one major symmetric conformation is observed, whereas in the F state, three asymmetric conformations are present. **B** Representative snapshots of the pore pentagon captured in each macrostate at each local free energy minimum.

The vertices of the pentagon correspond to the positions of the 9′ $C_\alpha$ atoms. The thickness of each pentagon indicates the relative population of that conformation within the corresponding macrostate. **C** Distribution of edge lengths of the 9′ $C_\alpha$ pentagon in each macrostate. State C displays the narrowest and most symmetric hydrophobic gate within the pore, whereas states O and D feature the most expanded pore radius. State F and I exhibits the most asymmetric pore profile. The distribution is weighted by the MSM weights.

PNU[16]. A total of 277 microseconds of simulations were performed for the CHOL system and analyzed as for the apo system, including the identification of five metastable states (C, F, O, D, I) (Supplementary Fig. 11). Similar variations in membrane thickness and local conformation were observed as in the apo system (Supplementary Fig. 12, Supplementary Fig. 13).

The resulting free-energy surface exhibited significant differences compared to the apo system. While state C still occupied the deepest basin, the desensitized State D was considerably stabilized in the presence of cholesterol (Fig. 6B) and the transition time from state O to state F remained around 100 μs (Fig. 6C, D). Notably, the energy barriers between states F, O, and D were flattened, leading to a faster transition from state O to state D (200 μs) compared to the apo system (1.8 ms). These findings indicate that cholesterol not only stabilizes the desensitized state but also facilitates the desensitization transition.

CHOL exhibited differential binding across five metastable states (Supplementary Fig. 10B). There was a more pronounced preference for binding at its assumed intersubunit sites (Fig. 6A, E, F, Supplementary Fig. 10A) in the desensitized state compared to the other states. In state D, CHOL interacted with M253 (Supplementary Fig. 10B), a residue previously shown to influence PNU binding in both experimental studies and simulations[16,44]. In state C, CHOL superficially bound at the periphery of the M1 and M3 helices; in state I and state F it shifted asymmetrically into the intersubunit pocket between these helices (Fig. 6E, F) from time to time. CHOL rarely accessed this binding site in states O.

## Discussion

Symmetry is a pervasive phenomenon in biological systems[45]. Comparable principles informed by physics have been employed in the modeling of intricate kinetics in dynamical systems, e.g., in fluid dynamics and control engineering[46,47]. These principles aim to extract coherent structures from high-dimensional observable time-series data by imposing specific constraints. The incorporation of such information can significantly improve data-driven approaches that are susceptible to noise and data size, and work around inaccuracies. As shown in this study, explicitly accounting for symmetry in MSMs by introducing SymTICA, we are both able to characterize the full gating cycle of the α7 nAChRs from molecular simulations with kinetics up to millisecond scale, but it also provides valuable insights into symmetry aspects of the transitions.

The relationship between different subunits and their transitions can be analyzed by examining the correlation among subIC values. If each subsystem operates independently, there should be no correlation between their subICs. For the α7 receptor, our findings showed a significant correlation between subIC1 and subIC2 across subunits, with Pearson correlation values of 0.85 for subIC1 and 0.64 for subIC2. This indicates a high degree of coupling in the gating processes of the receptors across the subunits, aligning with the receptor's allosteric characteristics[48]. Additional statistical models can be applied to further understand the strength of coupling between different subunits and how multiple subunits work together to achieve the gating process.

Recent advances in the decomposition of large molecular systems into independent subdomains have shown promise in unraveling the dynamics of complex systems[49–51]. A prerequisite for such decomposition is that each subsystem should ideally be weakly coupled or completely decoupled, displaying Markovian properties on their own. In contrast, SymTICA in this study takes a different approach by not assuming independence among the subsystems. Instead, it focuses on symmetric systems where the global dynamics rely on asymmetric pathways between states. This approach avoids the need to describe the transitions from each asymmetric pathway, which would require an exponential number of microstates. Nonetheless, it faces challenges in

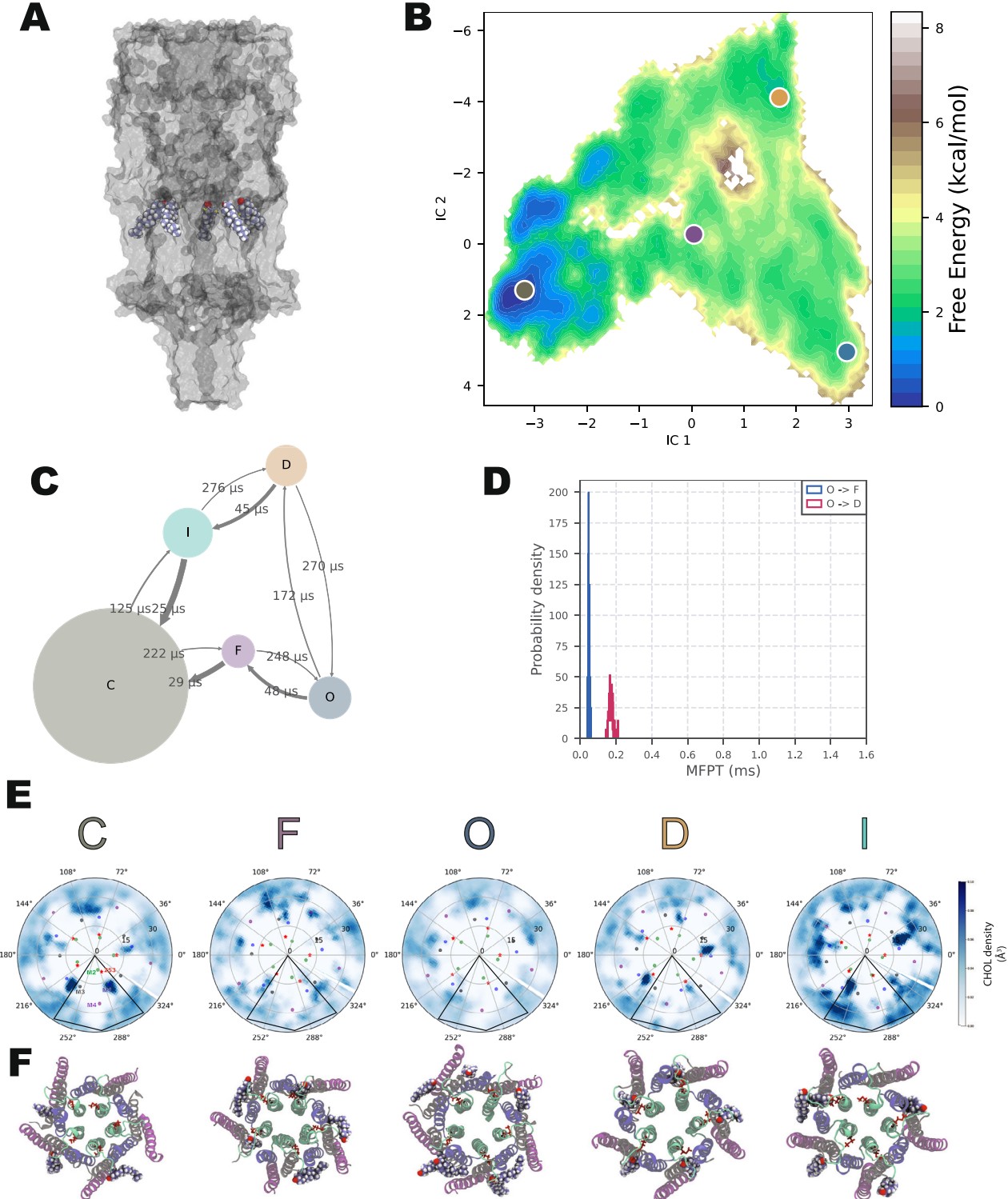

**Fig. 6 | Cholesterol stabilizes the desensitized state. A** Representative snapshot of the cholesterol binding site, with cholesterol shown as VDW spheres. **B** Free energy landscape of the CHOL system projected onto IC1-IC2 space. **C** Mean first passage times (indicated by arrows) between consecutive macrostates, with circle size indicating the relative stability of each macrostate. For simplicity, only passages along the predominant gating cycle are shown; for quantification of all transitions, see (Supplementary Fig. 8). **D** Histogram of mean first passage times between states O and F (blue) and between states O and D (orange), estimated from Bayesian MSM (100 samples). **E** Cholesterol binding profiles are generated for each macrostate. The histogram is plotted on a cylindrical axis, with the pore axis aligned with the z-axis. The sliced density around M253 (±1 Å along the z-axis) is displayed as a heatmap. Each subunit is represented by a skewed rectangle, the center of mass of each helix is depicted as a dot, and M253 is marked with a star. **F** Cholesterol binding snapshots for every macrostate. Cholesterols located within a 10 Å radius of M253 are represented as VDW balls, while the TMD helices are color-coded as follows: M1 (blue), M2 (green), M3 (gray), M4 (purple). M253 is depicted in red as licorice.

constructing a global MSM capable of describing all kinetics during gating. Future efforts should aim to integrate these approaches in order to enhance the construction of a more comprehensive MSM model. Deep learning frameworks have demonstrated promising capabilities in incorporating constraints, such as utilizing a matrix with a specific manifold[52], during the training process to effectively optimize solutions. Guided by the variational approach for Markov processes (VAMP), neural networks, such as VAMPNET[53], have been trained to maximize VAMP scores, enabling the capture of slow and nonlinear kinetics. We anticipate the principles employed in this study can also be applied as constraints during training in VAMPNET, thereby enhancing its performance while effectively capturing symmetry.

We have successfully constructed an MSM that establishes connections between three functional states of the $\alpha$ 7 nicotinic receptor: closed, open, and desensitized. Furthermore, this allowed us to identify two additional metastable states, which we suggest correspond to the flipped[12,21,22] (transitioning between closed and open) and intermediate (transitioning between closed and desensitized) functional states. Notably, the transition between the open and flipped states exhibits the characteristic nachschlag shuttings observed in single-channel recording studies[19,20], a phenomenon that is shared across members of the nicotinic receptor family. The congruence between transition times quantified in simulations without agonists, and those observed in electrophysiology experiments conducted with varying agonist concentrations support the hypothesis that the ancestral nature of spontaneous flipping is independent of agonism[23,41]. Nonetheless, because of the fleeting nature of these intermediate states, it is nearly impossible to structurally depict them e.g., with cryo-EM techniques. It is also important to acknowledge that the intermediate states observed in these simulations may not necessarily fully align with the sole flipped state responsible for the electrophysiology recordings.

Our study also offers direct structural models for asymmetric states during the gating process. We observed that the closed conformation displays lower flexibility compared to the open and desensitized states; higher flexibility in the latter states could facilitate ion permeation. Additionally, both intermediate states exhibit predominantly asymmetric configurations. Corresponding phenomena of asymmetric opening are not only evident in heteropentameric channels[54,55] but also in homopentamer forms[56]. This observation may be correlated with the fact that only one agonist molecule is required to initiate the opening of the $\alpha$ 7 channel[57].

Lipids serve as important endogenous modulators for membrane proteins[24,58,59]. Expanding upon our previous hypothesis[16], we investigated the role of cholesterol in the modulation of the desensitized state. Our MSM analysis supports the notion that cholesterol exhibits a differential binding profile, effectively stabilizing the desensitized state. Additionally, we observed that cholesterol facilitates the transition from the open to the desensitized state, potentially by introducing coupling between different domains in the transmembrane region. Previous studies examining the effect of cholesterol modulation on other subtypes of nAChRs have proposed similar mechanisms, wherein the absence of cholesterol leads to an uncoupled state, rendering the channel unable to function[24]. Another study highlighted the role of cholesterol in the $\alpha$7 subtype by modifying desensitization kinetics, possibly by disrupting lipid rafts[60]. Our findings suggest that cholesterol may also modulate channel function through direct binding, further emphasizing its impact on nAChR dynamics.

MSMs like the ones reported here can serve as powerful frameworks to extract representative snapshots from simulations. By leveraging the Boltzmann weights, they enable the generation of structural ensembles that can be utilized in drug design studies, for example by docking novel compounds to selectively stabilize specific states[61], which could also include stabilization of states not yet determined experimentally. Additionally, type II PAMs have been shown to bind to the intersubunit site−overlapping the binding site of cholesterol[16].

This implies that novel modulators could selectively stabilize a specific functional state by targeting residues that interact uniquely with cholesterol in that state (Supplementary Fig. 10B). We hope that future work can effectively harness these approaches to design improved modulators to enable more detailed pharmaceutical profiles for nAChRs.

## Methods

### Initial pathway generation
Six pathways were generated using Climber v. 1.0[36] by conducting forward and reverse interpolations between closed (PDB ID 7KOO), open (PDB ID 7KOX), and desensitized (PDB ID 7KOQ) states[14]. A total of 150 seeds were strategically chosen to provide a sparse sampling across the gating space. It is important to note that the initial pathways only need to adequately cover the necessary range, as we rely on unrestrained simulations for sampling and the MSMs to connect the states and modify the path.

### Molecular dynamics simulations
Simulation parameters are summarized in Supplementary Information. All seeds were inserted into 1-palmitoyl-2-oleoylphosphatidylcholine (POPC) lipid bilayers using MemProtMD[62]. They were then solvated with water (TIP3P[63]) and supplemented with 0.15 M NaCl to simulate physiological conditions. In the case of the cholesterol (CHOL) system, five cholesterol molecules were initially placed within the intersubunit binding sites for all seeds. Subsequently, seeds were embedded into bilayers composed of 33% cholesterol and 67% POPC, which mimics relevant physiological conditions.

For the simulations, ~400 POPC lipids were utilized for the apo system, while a combination of 300 POPC lipids and 150 cholesterol molecules were employed for the CHOL system. The simulation boxes had dimensions of $12 \times 12 \times 18$ nm³. To describe the system, the CHARMM36m forcefield (July 2020 version)[64] was employed. Atomistic forcefields are often preferred over coarse-grained to capture detailed conformational dynamics; for this protocol, the accuracy of the lipid parameters has also been previously validated[65].

To ensure system stability, relaxation was performed in several steps. First, energy minimization was performed using the steepest descent algorithm for 5000 steps. Subsequently, a three-stage NPT relaxation was carried out, with each stage lasting 10 ns. Initially, restraints were placed on all heavy atoms, followed by only backbone atoms, and finally only on C-$\alpha$ atoms of the protein. The docked cholesterol molecules were also restrained during the first two rounds of relaxation to maintain their positions, but unrestrained in the final stage.

To maintain the temperature at 300 K independently for the protein, lipid, and solvent components, the v-rescale thermostat[66] was employed. Additionally, the c-rescale barostat[67] was utilized to semi-isotropically maintain the pressure at 1 bar. Bond lengths were constrained using the LINCS[68] algorithm.

For production simulations, all restraints were removed, and systems simulated for 1–2 μs using GROMACS-2021[69], while employing the same thermostats and barostats as during equilibration. To enhance sampling along pathways between the closed and open states in the apo system, an additional 50 simulations were performed. These simulations were conducted along the pathway to improve conformational space coverage in less sampled regions. In total, the apo system underwent 200 μs of production simulations, while the CHOL system was simulated for 277 μs. These production simulations provided the necessary data for subsequent analyses, with data collection occurring at 1 ns intervals.

### Feature extraction
For the initial model from each seed, including the starting models, a contact map was constructed by considering the interactions between

all $C_\alpha$ atoms. A distance cutoff of 1 nm was applied to determine if two $C_\alpha$ atoms were in contact.

Contacts with a peak-to-peak value (range) versus mean ratio of less than 0.2 were deemed insignificant and removed from the analysis. This helps filter out contacts that do not exhibit substantial variation in their interaction. After filtering, a total of 1610 significant contacts remained. These contacts were organized into blocks to maintain invariance to the permutation of subunits (see next section). The block arrangement ensures that the order of contacts remains the same regardless of how the subunits are arranged. For example, the first block of contacts would encompass both intrasubunit contacts within subunit A and intersubunit contacts between subunits A and B. Reciprocal transformations were performed on all distances prior to featurization. This approach of using inverse distances makes the related tICAs less prone to domination by rare large fluctuations.

## Symmetry and augmentation in feature space

The $\alpha 7$ nicotinic receptors are homopentameric proteins composed of five identical subunits arranged in C5 symmetry around the central pore. This arrangement gives rise to rotational invariance, allowing us to understand the protein dynamics as a combination of five coupled subsystems.

For example, a simplified two-state (C, O) model can be used to describe the conformational states of the subsystems. The model can be represented as follows: $(C, C, C, C, C) \rightleftharpoons (C, C, C, C, O) \rightleftharpoons (C, C, C, O, O) \rightleftharpoons (C, C, O, O, O) \rightleftharpoons (C, O, O, O, O) \rightleftharpoons (O, O, O, O, O)$. It is important to note that within this model, there exist various degenerate asymmetric states, including configurations like (C, C, C, C, O) and (C, C, C, O, C), which share identical energy levels. Additionally, special semi-degenerate states, for instance (C, C, C, O, O), are also present, which can manifest in two distinct arrangements: (C, C, C, O, O) and (C, C, O, C, O). In constructing the model's features, we grouped such semi-degenerate states under a single category, although theoretically, they can be differentiated afterward based on their unique sequential subsystem combinations.

Considering the feature space, the observables that describe the dynamics of the system exhibit equivariance with respect to the symmetry group Z5 $\langle r | r^5 = e \rangle$. In other words, under the group actions, such as $\mathcal{F}_{perm} = \{f : r_p \circ f = f(x_A, x_B, x_C, x_D, x_E) = f(x_B, x_C, x_D, x_E, x_A)\}$, the Koopman propagation ($\mathcal{K}$) of general observable functions ($f$) commutes[46]:

$$r_p \circ (\mathcal{K}f)(x) = \mathcal{K}(r_p \circ f)(x) \tag{1}$$

The simulation sampling, thus, can be augmented by applying permutations to the original feature array from simulations when they are grouped based on the system's symmetry, which effectively improves the simulation sampling fivefold. Note that features in this system are organized into five subsystem blocks, rather than subunit blocks. The first block encompasses contacts within subunit 1, as well as contacts between subunit 1 and other subunits (primarily subunit 2) taking care to avoid double counting of the contacts.

## Symmetry-adapted time-lagged independent component analysis (SymTICA)

Time-lagged independent component analysis (TICA) is a dimensionality reduction technique utilized to extract the slowest dynamics present in a given feature space[37]. It operates on a sequence of time series data denoted as $\mathbf{X_t} = \{x_1, x_2, \ldots, x_N\}_t$ and computes the mean-free covariance matrix $C_0$ and the time-lagged covariance matrix $C_\tau$ given a specific lag time $\tau$. Under the assumption of reversible dynamics in TICA, the symmetry of $C_\tau$ is enforced numerically during the estimation process. The symmetric Koopman matrix, which encodes the kinetic information of the system, is then obtained. This matrix is

further decomposed into a spectrum of eigenvalues $\lambda$ and corresponding eigenvectors $v$. These eigenvalues and eigenvectors provide insights into the dominant slow modes or collective motions of the system, allowing for a reduced-dimensional representation of the dynamics.

In the presence of symmetry within the dataset, the covariance matrix of the Koopman matrix exhibits a distinct block form. For instance, in the case of 5-fold symmetry, the covariance matrix ($N \times N$) can be represented as:

$$\begin{bmatrix} \mathbf{C}_d & \mathbf{C}_{o1} & \mathbf{C}_{o2} & \mathbf{C}_{o2}^\top & \mathbf{C}_{o1}^\top \\ \mathbf{C}_{o1}^\top & \mathbf{C}_d & \mathbf{C}_{o1} & \mathbf{C}_{o2} & \mathbf{C}_{o2}^\top \\ \mathbf{C}_{o2}^\top & \mathbf{C}_{o1}^\top & \mathbf{C}_d & \mathbf{C}_{o1} & \mathbf{C}_{o2} \\ \mathbf{C}_{o2} & \mathbf{C}_{o2}^\top & \mathbf{C}_{o1}^\top & \mathbf{C}_d & \mathbf{C}_{o1} \\ \mathbf{C}_{o1} & \mathbf{C}_{o2} & \mathbf{C}_{o2}^\top & \mathbf{C}_{o1}^\top & \mathbf{C}_d \end{bmatrix} \tag{2}$$

Here, $\mathbf{C}_d$ ($N/5 \times N/5$) represents the diagonal block of the covariance matrix; $\mathbf{C}_{o1}$ ($N/5 \times N/5$) represents the off-diagonal block of the covariance matrix between features in subsystem i and i+1; $\mathbf{C}_{o2}$ ($N/5 \times N/5$) represents the off-diagonal block of the covariance matrix between features in subsystem i and i + 2.

Similarly, the Koopman matrix ($N \times N$) follows the same block structure, and it can be expressed as:

$$\begin{bmatrix} \mathbf{K}_d & \mathbf{K}_{o1} & \mathbf{K}_{o2} & \mathbf{K}_{o2}^\top & \mathbf{K}_{o1}^\top \\ \mathbf{K}_{o1}^\top & \mathbf{K}_d & \mathbf{K}_{o1} & \mathbf{K}_{o2} & \mathbf{K}_{o2}^\top \\ \mathbf{K}_{o2}^\top & \mathbf{K}_{o1}^\top & \mathbf{K}_d & \mathbf{K}_{o1} & \mathbf{K}_{o2} \\ \mathbf{K}_{o2} & \mathbf{K}_{o2}^\top & \mathbf{K}_{o1}^\top & \mathbf{K}_d & \mathbf{K}_{o1} \\ \mathbf{K}_{o1} & \mathbf{K}_{o2} & \mathbf{K}_{o2}^\top & \mathbf{K}_{o1}^\top & \mathbf{K}_d \end{bmatrix} \tag{3}$$

As a result, when solving the eigenvalue problem of the Koopman matrix $\mathbf{K}v = \lambda \mathbf{v}$, the N-dimensional eigenvector $\mathbf{v}$ can be decomposed as a concatenation of $v_\mathbf{A}$, $v_\mathbf{B}$, $v_\mathbf{C}$, $v_\mathbf{D}$, $v_\mathbf{E}$ ($N/5$-dimensional). The first component of the equation becomes $\mathbf{K}_d \cdot v_\mathbf{A} + \mathbf{K}_{o1} \cdot v_\mathbf{B} + \mathbf{K}_{o2} \cdot v_\mathbf{C} + \mathbf{K}_{o2}^\top \cdot v_\mathbf{D} + \mathbf{K}_{o1}^\top \cdot v_\mathbf{E} = \lambda v_\mathbf{A}$.

To enhance data efficiency and reduce estimation error in a large kinetic model (see the toy model system in Supplementary Information), we aim to understand the system dynamics under degeneracy. This involves seeking a symmetry-adapted mapping that remains invariant under the symmetry group. In this case, we have $v_\mathbf{A} = v_\mathbf{B} = v_\mathbf{C} = v_\mathbf{D} = v_\mathbf{E}$, and the eigenvalue problem can be simplified to $\mathbf{K}_{sum} v_\mathbf{A} = \lambda v_\mathbf{A}$, where $\mathbf{K}_{sum} = \mathbf{K}_d + \mathbf{K}_{o1} + \mathbf{K}_{o2} + \mathbf{K}_{o2}^\top + \mathbf{K}_{o1}^\top$.

The features $\mathbf{x_i}$ can be projected onto the dominant eigenspace (independent components, ICs) by summing over all five subsystems (subICs):

$$\mathbf{x_i} = \begin{bmatrix} \mathbf{x_{iA}} \\ \mathbf{x_{iB}} \\ \mathbf{x_{iC}} \\ \mathbf{x_{iD}} \\ \mathbf{x_{iE}} \end{bmatrix}, \quad \mathbf{subICs} = \begin{bmatrix} \mathbf{v_{A \cdot xiA}} \\ \mathbf{v_{A \cdot xiB}} \\ \mathbf{v_{A \cdot xiC}} \\ \mathbf{v_{A \cdot xiD}} \\ \mathbf{v_{A \cdot xiE}} \end{bmatrix}, \tag{4}$$

$$\mathbf{ICs} = \mathbf{v_A} \cdot x_{\mathbf{iA}} + \mathbf{v_A} \cdot x_{\mathbf{iB}} + \mathbf{v_A} \cdot x_{\mathbf{iC}} + \mathbf{v_A} \cdot x_{\mathbf{iD}} + \mathbf{v_A} \cdot x_{\mathbf{iE}} \tag{5}$$

Meanwhile, symmetry can be evaluated by analyzing the differences between each subICs, such as the standard deviation. Alternatively, symmetry-aware MDS[70] can be employed. Here, the distance matrix between samples in each macrostate was calculated from the minimum Euclidean distances of picked subIC(s) across all five possible permutations. It ensures that similar asymmetric conformations are embedded similarly, regardless of which subsystem differs. The distance matrix is then decomposed into a low-dimensional

representation using classical MDS. The symmetry-aware MDS was performed using the scikit-learn package[71] and is available in the Zenodo repository[72].

**Algorithm 1.** Get distance between two samples for five-fold symmetric systems

$S_i \leftarrow (S_{iA}, S_{iB}, S_{iC}, S_{iD}, S_{iE})$
$S_j \leftarrow (S_{jA}, S_{jB}, S_{jC}, S_{jD}, S_{jE})$
$D \leftarrow inf$
**for** $i \leftarrow 1$ **to** 5 **do**
    $S_j \leftarrow \text{roll}(S_j, 1)$
    $D \leftarrow \min(D, \text{Euclidean}(S_i, S_j))$
**end for**
**return** $D$

### Markov state model

The SymTICA analysis was conducted using a lag time of 50 ns, and the first two independent components (ICs) that separate the three starting states were selected. Other faster dimensions with a single Gaussian shape distribution in their sampling were excluded from further analysis[73].

After cross-validation based on the VAMP-2 score[74], 1000 microstates were clustered using the k-means algorithm[75] with k-means++[76] initialization, implemented in the Deeptime package[77]. Bayesian MSMs[78] were estimated from the discrete microstate trajectories with a lag time of 100 ns after the Markovian property was validated using the corresponding implied timescales[38] ranging from 50 ns to 500 ns.

Subsequently, a five-state coarse-grained MSM was constructed using the PCCA+ algorithm[79,80], which partitions the microstates into kinetically distinct sets. The MSM was validated using the Chapman-Kolmogorov test[38].

All MSM analyses were performed using either the Deeptime[77] or PyEMMA[81] packages, and the results were visualized using seaborn[82], Matplotlib[83], or prettypyplot[84].

### Relevant feature analysis

The contact features, as well as other geometric features, were computed using custom scripts available at the GitHub repository: https://github.com/yuxuanzhuang/msm_a7_nachrs; the scripts utilized MDAnalysis v. 2.8.0[85] and ENPMDA[72].

To calculate membrane thickness, the lipophilic package[86] was employed. The thickness was determined by selecting phosphorus (P) atoms from POPC molecules and measuring the distance between the upper and lower leaflets of the membrane.

Pore hydration around residue 9′ or 2′ was defined as the number of water molecules within a cylindrical region centered at the specific residue and spanned ± 2 Å along the pore axis. The tilts of the TMD helices were determined using the HELANAL algorithm[87]. The tilt angle between each helix and the membrane normal was calculated, providing information about the orientation of the helices relative to the membrane.

All distributions plotted were reweighted by the stationary probability of each microstate.

### Cholesterol density calculation with symmetry-corrected alignment

To evaluate the cholesterol density around the protein, conventionally, a single reference structure is used to first align all the trajectories based on the minimum root-mean-square deviation (RMSD) metric. The density is calculated with a grid-based approach. However, this approach is not suitable for asymmetric proteins from multiple trajectories because different trajectories may sample different asymmetric conformations but still belong to the same state. Therefore, we developed a symmetry-corrected alignment method to align the trajectories based on the symmetry of the protein. This method is an extension of the RMSD-based structure alignment algorithm implemented in MDAnalysis[85,88,89] to obtain the minimum RMSD value for each permutation and the corresponding rotation matrix.

### Reporting summary

Further information on research design is available in the Nature Portfolio Reporting Summary linked to this article.

## Data availability

Representative snapshots from each microstate and final MSM models have been deposited on Zenodo at https://zenodo.org/records/11117001. The source data underlying Figs. 1D–E, 2B, D, E, 3A, B, 4B–G, 5A–C, 6B, D, and Supplementary Figures are provided as a Source Data file. Structures with accession codes 7KOX, 7KOO, and 7KOQ were used as starting models. Source data are provided with this paper.

## Code availability

The scripts to generate figures can be found on GitHub at https://github.com/yuxuanzhuang/msm_a7_nachrs. The implementation of SymTICA can be found at https://github.com/yuxuanzhuang/sym_msm. Code and scripts that reproduce the results are also available on Zenodo at https://zenodo.org/records/11117001[72].

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

## Acknowledgements

This work was supported by grants from the Knut and Alice Wallenberg Foundation, the Swedish Research Council (2019-02433, 2021-05806), the Swedish e-Science Research Centre, and the BioExcel Center of Excellence (101093290). Computational resources were provided by the National Academic Infrastructure for Supercomputing in Sweden (NAISS/SNIC 2022/3-40, 2022/21-16), the European High-Performance Computing Joint Undertaking (EuroHPC JU), and the Swiss National Supercomputing Centre (CSCS) through the Partnership for Advanced Computing in Europe (PRACE). The authors are grateful to Prof. Ryan E. Hibbs and Prof. Lucie Delemotte for helpful feedback and discussions.

## Author contributions

Y.Z., R.J.H., and E.L. designed research; Y.Z. performed research and analyzed data; Y.Z., R.J.H., and E.L. wrote the paper; Y.Z., R.J.H., and E.L. revised the paper.

## Funding

## Competing interests

The authors declare no competing interests.
