## [Peer Review File · Nature Communications]

Symmetry-adapted Markov state models of closing, opening, and desensitizing in $\alpha 7$ nicotinic acetylcholine receptorsREVIEWER COMMENTS

Reviewer #1 (Remarks to the Author):

In their article „Symmetry-adapted Markov state models of closing, opening, and desensitizing in a7 nicotinic acetylcholine receptors“, Zhuang et al. present extensive MD simulations on a nicotinic acetylcholine receptor, which is a membrane ion channel. Furthermore, they present a new approach to symmetrize Markov state models to take into account symmetries in homo-oligomeric protein complexes.

Generally, I recommend publication of the presented manuscript in Nat Commun: the proposed symmetrization of MSMs is of methodological interest. From the side of biophysics, the investigated biophysical system is an interesting and representative template for symmetric ion channels, and the amount of accumulated simulation data is significant. However, in its current state, the description of the symmetrization and the discussion of its consequences need to be extended, and the trajectory analysis holds more potential to discuss allosteric coupling between the subunits as well as the basis for the apparent sequential order of conformational changes. I therefore recommend the following major revisions:

-General remarks:

As far as I understand the presented symmetrization approach of MSMs, the target of the approach is to correct for the rotational symmetry, thus improving the number of sampled transitions and focusing on the subunits themselves rather than the potential coupling between subunits. However, what I am missing is a discussion of this (potential) allosteric coupling between subunits. As the authors already pointed out, the comparison between projections along ICs and subICs hints at an intrinsic asymmetry – may that be related to allosteric coupling between subunits, of possibly even cooperative transitions?

Furthermore, the authors present free energy landscapes based on only the first two ICs. Fig. S6 however shows that IC3 can be active on the same time scale as IC2. What process does IC3 represent? How do projections along IC1-IC3 and IC2-IC3 look like, and what states do these projections resolve? The free energy landscapes as in, e.g., Fig. 2, are apparently deceiving, as based on the presented barriers, states I and F should exchange via a common barrier, yet they apparently do not according to MSMs.

How many transitions were observed exactly between the individual states? This information is of importance to judge on the quality of the MSMs.

The authors write about a sequential order of conformational changes that appears based on an analysis of the MSMs. This seems contradictory: sequential orders of events imply the existence of Memory, which is removed in the generation of MSMs. How can conformational changes then be sequential? What signatures of sequential changes do, e.g., the MSMs in Figs. 2 and 6 display?

-detailed remarks:

Introduction:

p. 3, 3rd paragraph:

The authors write here about a sequential order of conformational changes that appears based on an analysis of the MSMs – see my general remark above.

Methods:

p. 4, line 161: "reciprocal transformations were performed on the contact distances": does that mean that the analysis was based on so-called "inverse distances"? What was the reasoning behind this approach?

p. 4, 2nd-to-last paragraph: this point may be unrelated to the data evaluation of the simulation system itself, but the presented simplified two-state model misses a state (C,O,C,O,C) – in the current model, the mechanism looks completely sequential. I assume that the simulation data analysis routine by its blocking of contact distances and preparation of virtual trajectory copies is not affected by presuming an underlying transition mechanism?

p. 6, top: the explanation of symmetry-aware multidimensional scaling is a bit short – it would be good to explain what information this method yields (see as well my comment below).

Results:

Fig. 3: The insets with structural overview of water molecules entering the channel are very small. Please increase them in size.

pp. 12-13 and Fig. 5: the part on "Symmetry in gating transitions" is much too short. Please explain better: what does MDS exactly yield as information? Maybe a slight extension of the explanation in the caption of Fig. 5 would be enough. Furthermore, I do not understand the 9' C_α pentagon plots, please explain what the reader is expected to see here.

Supplementary information:

Fig. S7: The CK-tests in panels B and D are too small to evaluate the quality of the MSMs Please enlarge or (even better) turn both subpanels into separate SI Figures.

Reviewer #2 (Remarks to the Author):

The authors present a computational study of the kinetic behaviour of the homomeric Alpha-7 nicotinic receptor, both in the presence and in the absence of cholesterol.

The simulations — which amount to almost half a millisecond — were initiated by three experimentally available structures representing the biologically relevant conformations and were used to obtain Markov State kinetic model after accounting for the 5-fold symmetry of the protein.

The system setup, execution of the simulations, analysis of convergence are rigorous and comply with the best practices of the field.

While the tools introduced here for accounting for the symmetry of the system to enhance robustness and statistical significance are intriguing and potentially valuable to the broader community focused on computational modeling of symmetric systems, the methodological advancement by itself appears better aligned with a journal catering to a more specialized readership.

The clarity and details of the methodological description, moreover, should be improved.

The authors offer no validation of their computational conclusions. The biological and pharmacological relevance of the work could be improved by supplying, or at least proposing and motivating,

experimental validations of the mechanistic findings. Are there specific mutations that would modulate the kinetic behaviors in an experimentally observable manner? (faster desensitizing, shorter openings, etc.) The authors also note that the insight into the dynamics of the receptor could be used to “design improved modulators”. What would be the principles of such design campaign? Do the finding point to novel strategies to target this important receptor?

Specific comments:

1. The details of the methods could be improved and made more clear in several passages.

1.1. Page 4 line 169 “Multiple semi-degenerate asymmetric states” what does “semi-degenerate” mean?

1.2. Page 192-193 The description of the block structure of the correlation matrix on Pag 5, lines 192-195 is a bit confusing to me. As far as I understand the correlation between —say— two (subunit 1-subunit 2) contacts is in Cd, while Co1 would contain a correlation between a (subunit 1-subunit 2) contact and a (subunit 2-subunit 3) contact. So the description “Co1 represents the off-diagonal block of the covariance matrix between features in subsystem i and i+1” is not clear.

1.3 Pag 5, line 198. Does $[v_A, v_B, v_C, v_D, v_E]$ indicate the direct sum of the block vectors? The transformed eigenvector equation is not clear: isn't the dimension of v five times the one of v_A ? Did you mean to write λv_A and not λv ? i.e., $K_d \cdot v_A + K_{o1} \cdot v_B + K_{o2} \cdot v_C + K_{o1}^T \cdot v_D + K_{o2}^T \cdot v_E = \lambda v_A$. Also the equation on line 204 seems to have an extra prime. Did you mean $K' v_A = \lambda v_A$? Also, it would help to provide a clear (formulas are usually better than a word description) definition of how the regular TICA projections are obtained from the “sub-independent components”.

1.4 Page 6, 208 “symmetry-aware MDS” I could not find any description of this method in the literature, so more details should be presented. The authors write “[...] the adjacency matrix is calculated from the minimum Euclidean distances among all permutations, allowing for the estimation of symmetry”. First of all is not clear what “Euclidean distances” means. Did the authors calculate the distance in $3N$ dimensions where N is the total number of atoms in the receptor? or rather in the feature-space used for the MSM, i.e. inter-atomic distances? Furthermore, “adjacency matrix” is usually reserved for matrices with binary entries (contact-no contact), while MDS embeds distances, so I'd use “distance matrix”. Can the authors comment on how calculating the distance as the minimum over subunit permutations helps estimate the symmetry of the system?

2. Figure 2 and figure 6, panel D. I think the axis label is misleading (should be “first-passage-time”): there should be only one *mean* first-passage-time. Moreover, the MSM is based on microstates, whereas the plot shows the transitions between coarse-grained states. It would be clearer if the authors clarified in detail how the histogram was “estimated from the MSM”. Are the results in the histogram (panel D) in agreement with panel C? For instance, in fig 2C, O->D is 1.8 ms but there is no density in panel D beyond 1.6 ms.

3. Page 8, line 295 “when we projected existing 7 cryo-EM structures onto the IC space, one particular structure resolved in the presence of EVP-6124 and PNU fell into the basin corresponding to the F state” It would be interesting to see where the other structures (3?) end up

Reviewer #2 (Remarks on code availability):

I commend the authors for sharing their code. I am well aware of the amount of work it takes to take a functioning code and make it available and usable by other researchers.

I have browsed the code in the repositories, and could install the `sym_msm` library via pip.

However, the documentation for both repositories is still missing navigating the code without comments would take more effort that I can devote to this review, so I cannot comment on the reproducibility of the results.

Erik Lindahl
Professor of Biophysics

Stockholm, 5 May 2024

Dear Editor,

Thank you for the thoughtful consideration of our work. As demonstrated in the attached revision and outlined below, we believe we have responded to all concerns, including edits to the Introduction, Methods, Results, Discussion, Figures 2, 3, 5, 6, S1, S6–S10, S13, and new SI content Table S1 and Figure S11.

In this response, reviewer “recommendations for the authors” are gray (renumbered for clarity), and quotes from the revised manuscript are written in red.

Reviewer #1 (Remarks to the Author):

In their article „Symmetry-adapted Markov state models of closing, opening, and desensitizing in a7 nicotinic acetylcholine receptors”, Zhuang et al. present extensive MD simulations on a nicotinic acetylcholine receptor, which is a membrane ion channel. Furthermore, they present a new approach to symmetrize Markov state models to take into account symmetries in homo-oligomeric protein complexes.

Generally, I recommend publication of the presented manuscript in Nat Commun: the proposed symmetrization of MSMs is of methodological interest. From the side of biophysics, the investigated biophysical system is an interesting and representative template for symmetric ion channels, and the amount of accumulated simulation data is significant. However, in its current state, the description of the symmetrization and the discussion of its consequences need to be extended, and the trajectory analysis holds more potential to discuss allosteric coupling between the subunits as well as the basis for the apparent sequential order of conformational changes. I therefore recommend the following major revisions:

General remarks:

R1.C1. As far as I understand the presented symmetrization approach of MSMs, the target of the approach is to correct for the rotational symmetry, thus improving the number of sampled transitions and focusing on the subunits themselves rather than the potential coupling between subunits. However, what I am missing is a discussion of this (potential) allosteric coupling between subunits. As the authors already pointed out, the comparison between projections along ICs and subICs hints at an intrinsic asymmetry – may that be related to allosteric coupling between subunits, of possibly even cooperative transitions?

Thank you for your constructive feedback regarding the importance of exploring allosteric coupling and its impact on cooperativity between different subunits within our study! We recognize the significance of this aspect for a comprehensive understanding of the system's behavior and have taken steps to incorporate this perspective into our revised manuscript.

We first endeavored to develop a more rigorous framework for assessing allosteric coupling by assigning discrete states to each subsystem and deriving the global state from their combinations, but failed. Specifically, the combinatorial nature of this approach led to an impractical number of global states. For example, assigning ten possible states to each subsystem would result in 10^5 potential global states when disregarding degeneracy. Achieving convergence became computationally impossible with samples from simulations.

Department of Biochemistry and Biophysics

Stockholm University
Science for Life Laboratory
Box 1031
171 21 Solna, Sweden

Visiting address:
Science for Life Laboratory
Tomtebodavägen 23A
171 65 Solna, Sweden

Phone: +46-734618050
Cell: +46-734618050
Mail: erik.lindahl@scilifelab.se

In response to the comments, we have included a new paragraph in the Discussion section, detailing our efforts to investigate the correlation between subIC values and potential allosteric coupling, offering initial insights into how these interactions may influence the system's overall cooperativity:

p. 16 l. 437: **The relationship between different subunits and their transitions can be analyzed by examining the correlation among subIC values. If each subsystem operates independently, there should be no correlation between their subICs. For the $\alpha 7$ receptor, our findings showed a significant correlation between subIC1 and subIC2 across subunits, with Pearson correlation values of 0.85 for subIC1 and 0.64 for subIC2. This indicates a high degree of coupling in the gating processes of the receptors across the subunits, aligning with the receptor's allosteric characteristics (75). Additional statistical models can be applied to further understand the strength of coupling between different subunits and how multiple subunits work together to achieve the gating process.**

R1.C2. Furthermore, the authors present free energy landscapes based on only the first two ICs. Fig. S6 however shows that IC3 can be active on the same time scale as IC2. What process does IC3 represent? How do projections along IC1-IC3 and IC2-IC3 look like, and what states do these projections resolve?

Thank you for the suggestion to share more analyses of IC3. We chose to exclude IC3 from our central published analysis because it corresponds to a fairly flat free energy landscape that does not differentiate between any states, and appears to represent a common motion in simulations that is not state-dependent (Reviewer Figure 1). Including IC3 resulted in a less converged Markov State Model (MSM), likely capturing a slow yet uninteresting (from the state-transition perspective) motion.

Reviewer Figure 1. Free-energy surface of $\alpha 7$ -nAChR gating projected onto IC1-IC3 coordinates. Circles represent projections of closed (gray), open (blue), and desensitized (yellow) structures.

For transparency, we have added labels to IC1–6 in Supplementary Figure S6. We also now comment on this point in Results:

p. 9 l. 297: IC1 primarily captured the separation between the closed state and the open/desensitized states, while IC2 primarily captured the transition between the desensitized state and closed/open states (Figure 1D-E, Figure S6). No major state differentiation was apparent in projections along shorter-timescale components (IC3–6) (Figure S6).

Figure S6: Implied timescale of each independent component. (A, C) Implied timescale of each independent component for the apo and CHOL systems. (B, D) Histogram of each independent component for the apo and CHOL system. Structural models are projected onto the plot (grey: closed; blue: open; yellow: desensitized).

R1.C3. The free energy landscapes as in, e.g., Fig. 2, are apparently deceiving, as based on the presented barriers, states I and F should exchange via a common barrier, yet they apparently do not according to MSMs. How many transitions were observed exactly between the individual states? This information is of importance to judge on the quality of the MSMs.

We thank the reviewer for this helpful comment. In line with the recommendation, we have conducted an analysis on the number of transitions between all pairs of macrostates in a new **Supplementary Figure S11**. This investigation revealed numerous (more than 100) transitions from and to each macrostate, including the newly identified I and F macrostates, indicating that no single state is isolated within the gating cycle. We have modified the text accordingly:

p. 9 l. 309: The validity of the Markov State Model (MSM) is confirmed through the application of both the Chapman-Kolmogorov (CK) test [55] and the enumeration of transitions between states (Figure S7, Figure S8, Figure S9, Figure S11).

p. 10 Figure 2. (C) Mean first passage times (indicated by arrows) between consecutive macrostates, with circle size indicating the relative stability of each macrostate. For simplicity, only passages along the predominant gating cycle are shown; for quantification of all transitions, see Figure S11.

Figure S11: Transition counts in MSMs. Number of transitions between each macrostate for apo (left) and CHOL (right) systems, colored indigo–yellow with increasing number of transitions.

Note that we also estimated the errors associated with these transitions based on a Bayesian MSM in Supplementary Figure S10, providing a basis for our model's reliability.

R1.C4. The authors write about a sequential order of conformational changes that appears based on an analysis of the MSMs. This seems contradictory: sequential orders of events imply the existence of Memory, which is removed in the generation of MSMs. How can conformational changes then be sequential? What signatures of sequential changes do, e.g., the MSMs in Figs. 2 and 6 display?

The reference to "sequential conformational changes" in the context of Markov State Models (MSM) might indeed seem perplexing. The term "sequential order" is merely employed from the viewpoint of electrophysiology experiments, in which a channel is presumed to progress through a series of states: starting from a closed conformation, transitioning to an open state, moving into a desensitized state, and finally returning to a closed state. This presumed progression provided a sequential framework for the macrostates identified in our study, though of course kinetics of additional transitions are also available in the MSM.

Furthermore, we propose considering these sequential conformational changes in terms of conditional probabilities. For instance, given that the channel is in its open state, we can predict the likelihood of further or less tilting of the M2 helix (as detailed in our manuscript). This approach does not conflict with the nature of memorylessness in the MSM itself. To clarify these points, we have expanded the beginning of the Results section "Sequential conformational changes in gating":

p. 11 l. 349: **To comprehend the structural transitions that occur across the presumed gating cycle of the channel---from closed to open, to desensitized, and finally back to closed---as likely observed in electrophysiology experiments, we focused on this sequence of conformational changes and the corresponding dynamics.**

Detailed remarks:

Introduction:

R1.C5. p. 3, 3rd paragraph:

The authors write here about a sequential order of conformational changes that appears based on an analysis of the MSMs – see my general remark above.

Please see the response to R1.C4 above regarding the sense in which we refer to a sequential order of conformational changes, i.e. discrete states associated with one state or another in a presumed functionally relevant cycle. We have also modified the relevant sentence to clarify this point:

p. 3 l. 107: **By analyzing the MSMs we were able to map the free energy landscape of the entire functional cycle of the $\alpha 7$ nAChR, and in particular identify local structural transitions associated with the presumed functional gating cycle, generally extending from the ECD to the TMD through the coupling region.**

Methods:

R1.C6. p. 4, line 161: “reciprocal transformations were performed on the contact distances”: does that mean that the analysis was based on so-called “inverse distances”? What was the reasoning behind this approach?

We appreciate that the details and rationale for this approach were not clear. In principle, using inverse distances rather than distances between Ca atoms for featurization can reduce the influence of rare large-scale fluctuations on the characterization of tIC's. The impact of this strategy may have been limited by our initial trimming of distant pairs (≥ 1 nm); indeed, we found that inverse distances yielded similar mappings and VAMP2 scores for the $\alpha 7$ system, only marginally shortening the longest implied timescale (tIC1, from 20 μ s to 19 μ s, n.s.). To explain the rationale in principle, we have added a sentence to the “Feature extraction” subsection in Methods:

p. 4 l. 164: **Reciprocal transformations were performed on all distances prior to featurization. This approach of using inverse distances makes the related tICs less prone to domination by rare large fluctuations.**

R1.C7. p. 4, 2nd-to-last paragraph: this point may be unrelated to the data evaluation of the simulation system itself, but the presented simplified two-state model misses a state (C,O,C,O,C) – in the current model, the mechanism looks completely sequential. I assume that the simulation data analysis routine by its blocking of contact distances and preparation of virtual trajectory copies is not affected by presuming an underlying transition mechanism?

We thank the reviewer for this insightful comment. The routine for simulation analysis remained unchanged because we did not create the final Markov State Model (MSM) by combining five subsystem states. As described above in response to R1.C1, assembling it in this way became impractical due to limited sampling and the extensive variety of potential global states. It should be possible to extract such information, for example by analyzing sequential subIC values to determine the distribution of different asymmetric states. However, we found that the data from these analyses did not yield useful information, as they were predominantly noisy.

In the Methods section, we now provide additional details regarding special "semi-degenerate" states such as COCOC:

p. 4 l. 173: **It is important to note that within this model, there exist various degenerate asymmetric states, including configurations like (C, C, C, C, O) and (C, C, C, O, C), which share identical energy levels. Additionally, special semi-degenerate states, for instance (C, C, C, O, O), are also present, which can manifest in two distinct arrangements: (C, C, C, O, O) and (C, C, O, C, O). In constructing the model's features, we grouped such semi-degenerate states under a single category, although theoretically, they can be differentiated afterward based on their unique sequential subsystem combinations.**

R1.C8. p. 6, top: the explanation of symmetry-aware multidimensional scaling is a bit short – it would be good to explain what information this method yields (see as well my comment below).

In the example notebook on GitHub (https://github.com/yuxuanzhuang/sym_msm/blob/main/example/example_sym_mds.ipynb), we demonstrate that when introducing asymmetry to one of the subsystems in a toy Gaussian system, symmetry-aware multidimensional scaling (MDS) can accurately capture the scale of the asymmetry (Reviewer Figure 2).

Reviewer Figure 2. The symmetry-aware multidimensional scaling (MDS) analysis was applied to four synthetic datasets where the asymmetry was modified and controlled using an asymmetric scale. In the presence of asymmetry, the sampling deviates from the zero point and exhibits a wider distribution in the MDS mapping.

To clarify this topic, we have substantially expanded the explanation of MDS. Specifically, we have rewritten the Methods section for the symmetry-aware MDS methodology. We have also included a block of algorithmic explanation to clarify how distances between samples are calculated. It specifically computes the Euclidean distances using the array formed from the subIC values:

p. 6 l. 225: **Alternatively, symmetry-aware multidimensional scaling (MDS) [47] can be employed. Here, the distance matrix between samples in each macrostate was calculated from the minimum Euclidean distances of picked subIC(s) across all five possible permutations. It ensures that similar asymmetric conformations are embedded similarly, regardless of which subsystem differs. The distance matrix is then decomposed into a low-dimensional representation using classical MDS. The symmetry-aware MDS was performed using the scikit-learn package and is available in the GitHub repository (https://github.com/yuxuanzhuang/sym_msm).**

Results:

R1.C9. Fig. 3: The insets with structural overview of water molecules entering the channel are very small. Please increase them in size.

Thanks for the suggestion. We have increased the insets of snapshots in **Figure 3**.

R1.C10. pp. 12-13 and Fig. 5: the part on “Symmetry in gating transitions“ is much too short. Please explain better: what does MDS exactly yield as information? Maybe a slight extension of the explanation in the caption of Fig. 5 would be enough. Furthermore, I do not understand the 9' C_alpha pentagon plots, please explain what the reader is expected to see here.

With thanks for this comment, we have expanded the Methods section on MDS as described in response to R1C8, and have also enhanced the caption of Figure 5 to more clearly explain how MDS captures asymmetric states:

p. 14 Fig. 5: **Asymmetric intermediate states.** (A) Symmetry-aware multidimensional scaling (MDS) projected sub Independent Components (IC1) onto the MDS1-MDS2 space. The zero point represents perfect symmetry with mean value, while a shifted distribution suggests sampling of asymmetric conformations. For instance, in the C system, only one major symmetric conformation is observed, whereas in the F state, three asymmetric conformations are present. (B) Representative snapshots of the pore pentagon captured in each macrostate at each local free energy minimum. The vertices of the pentagon correspond to the positions of the 9' Ca atoms. The thickness of each pentagon indicates the relative population of that conformation within the corresponding macrostate. (C) Distribution of edge lengths of the 9' Ca pentagon in each macrostate. State C displays the narrowest and most symmetric hydrophobic gate within the pore, whereas states O and D feature the most expanded pore radius. State F and I exhibits the most asymmetric pore profile. The distribution is weighted by the MSM weights.

We have also added the following sentences to the Results paragraph describing the pentagon plots:

p. 13 l. 390: **Among these minimum-energy structures, we visualized differential pore symmetry via the pentagonal belt located at each channel's hydrophobic gate (Figure 5B).** The vertices of each pentagon correspond to Ca atoms of the 9' residues. In a symmetric conformation, the geometry would be regular; in an asymmetric pore, it would be more or less distorted.

Supplementary information:

R1.C11. Fig. S7: The CK-tests in panels B and D are too small to evaluate the quality of the MSMs Please enlarge or (even better) turn both subpanels into separate SI Figures.

Thanks for the suggestion! We have enlarged and turned both CK-tests into separate SI Figures (**S8 and S9**).

Reviewer #2 (Remarks to the Author):

The authors present a computational study of the kinetic behaviour of the homomeric Alpha-7 nicotinic receptor, both in the presence and in the absence of cholesterol.

The simulations — which amount to almost half a millisecond — were initiated by three experimentally available structures representing the biologically relevant conformations and were used to obtain Markov State kinetic model after accounting for the 5-fold symmetry of the protein.

The system setup, execution of the simulations, analysis of convergence are rigorous and comply with the best practices of the field.

While the tools introduced here for accounting for the symmetry of the system to enhance robustness and statistical significance are intriguing and potentially valuable to the broader community focused on computational modeling of symmetric systems, the methodological advancement by itself appears better aligned with a journal catering to a more specialized readership. The clarity and details of the methodological description, moreover, should be improved.

R2.C1. The authors offer no validation of their computational conclusions. The biological and pharmacological relevance of the work could be improved by supplying, or at least proposing and motivating, experimental validations of the mechanistic findings. Are there specific mutations that would modulate the kinetic behaviors in an experimentally observable manner? (faster desensitizing, shorter openings, etc.) The authors also note that the insight into the dynamics of the receptor could be used to “design improved modulators”. What would be the principles of such design campaign? Do the finding point to novel strategies to target this important receptor?

We believe that another potential strategy for designing modulators, in addition to the ensemble docking approach described in the Discussion section (p. 17 l. 489), could be to leverage the differential interactions of cholesterol with various states as a probe. This approach would aid in identifying potential interacting residues for novel modulators. Consequently, we have included data on leading contact frequencies as a new panel B in **Supplementary Figure S13**, and expanded the related Results and Discussion paragraphs to elaborate on this strategy:

p. 14 l. 418: **CHOL exhibited differential binding across five metastable states (Figure S13B. There was a more pronounced preference for binding at its assumed intersubunit sites (Figure 6A,E,F, Figure S13A) in the desensitized state compared to the other states. In state D, CHOL interacted with M253 (Figure S13B), a residue previously shown to influence PNU binding in both experimental studies and simulations [16, 72].**

p. 17 l. 499: **Additionally, type II positive allosteric modulators have been shown to bind to the intersubunit site---overlapping the binding site of cholesterol [16]. This implies that novel modulators could selectively stabilize a specific functional state by targeting residues that interact uniquely with cholesterol in that state (Figure S13B). We hope that future work can effectively harness these**

approaches to design improved modulators to enable more detailed pharmaceutical profiles for nAChRs.

Figure S13: Top 10 residues interacting with cholesterol in each macrostate, ordered by the greatest variance across five states.

Specific comments:

R2.C2. 1. The details of the methods could be improved and made more clear in several passages. Page 4 line 169 “Multiple semi-degenerate asymmetric states” what does “semi-degenerate” mean?

We appreciate the recommendations to enhance the specificity of our methods section. We have now added a supplementary summary of simulation conditions (Table S1), and expanded on our forcefield selection:

p. 3 l. 123: Simulation parameters are summarized in Table S1.

p. 3 l. 132: Atomistic forcefields are often preferred over coarse-grained to capture detailed conformational dynamics; for this protocol, the accuracy of the lipid parameters has also been previously validated [40].

The term "semi-degenerate states" refers to configurations such as (C, C, C, O, O) that can appear in different, distinct arrangements—namely, (C, C, C, O, O) and (C, C, O, C, O). To clarify, we have provided a more detailed explanation of these states within the **Methods section** of our manuscript (p. 4 l. 172), as detailed in response to R1C7.

R2.C3. Page 192-193 The description of the block structure of the correlation matrix on Pag 5, lines 192-195 is a bit confusing to me. As far as I understand the correlation between —say— two (subunit 1-subunit 2) contacts is in Cd, while Co1 would contain a correlation between a (subunit 1-subunit 2) contact and a (subunit 2-subunit 3) contact. So the description “Co1 represents the off-diagonal block of the covariance matrix between features in subsystem i and i+1” is not clear.

We agree that the method description here was not sufficiently clear. We have now added the following to Methods:

p. 5 l. 187: Note that features in this system are organized into five subsystem blocks, rather than subunit blocks. The first block encompasses contacts within subunit 1, as well as contacts between subunit 1 and other subunits (primarily subunit 2), taking care to avoid double counting of contacts.

Consequently, the correlation between features within a single subsystem, such as subsystem 1, appears in the first diagonal block; for subsystem 2, it appears in the second diagonal block. These blocks are identical due to the permutation invariance of the system. The first off-diagonal block, Co1, then represents the covariance of features between subsystem 1 and subsystem 2.

To enhance the visualization of the Koopman matrix, we have also included it for the toy system as **Supplementary Figure S1F** (previously Appendix figure 1, cited p. 6 l. 214). This addition makes it easier to observe that two diagonal blocks are identical. We can also tell that synthetic features 1 and 2 of subsystem 1 are highly correlated and play a crucial role in the first two implied count (IC) transformations. What SymTICA accomplishes is the eigenvalue decomposition of K_d , and using the leading eigenvectors to transform features across all subsystems.

R2.C4. Pag 5, line 198. Does $[v_A, v_B, v_C, v_D, v_E]$ indicate the direct sum of the block vectors? The transformed eigenvector equation is not clear: isn't the dimension of v five times the one of v_A ? Did you mean to write λv_A and not λv ? i.e., $K_d \cdot v_A + K_{o1} \cdot v_B + K_{o2} \cdot v_C + K_{o1}^T \cdot v_D + K_{o2}^T \cdot v_E = \lambda v_A$. Also the equation on line 204 seems to have an extra prime. Did you mean $K' v_A = \lambda v_A$? Also, it would help to provide a clear (formulas are usually better than a word description) definition of how the regular TICA projections are obtained from the “sub-independent components”.

Thank you for identifying these errors in our description! The notation $[v_A, v_B, v_C, v_D, v_E]$ represents the concatenation of five arrays, each from a different block. The reviewer is correct in noting that it should be λv_A instead of λv_B . Additionally, we have replaced the prime notation with K_{sum} to clarify that the Koopman matrix used for the eigendecomposition in symTICA is the sum of the five block matrices.

In regular TICA, the vectors $v_A, v_B, v_C, v_D,$ and v_E are not required to be identical, which results in additional solutions for the eigenvalues, seen in the 3rd, 4th, and 5th eigenvectors. These eigenvectors might indicate interactions/couplings between subsystems and can distinguish asymmetric states like (C,C,C,C,O) and (C,C,C,O,C). We don't think it's possible to capture all the full eigenvectors from subICs because information related to coupling between e.g. K_d and K_o are lost when they are summed. We have also included the formulas to better illustrate how the projection is performed from sub-independent components in.

In light of these comments, we have accordingly updated the **Methods section** regarding SymTICA in our manuscript (p. 5 l. 209):

209 Similarly, the Koopman matrix ($N \times N$) follows the same block structure, and it can be expressed
210 as:

$$\begin{bmatrix} K_d & K_{o1} & K_{o2} & K_{o2}^T & K_{o1}^T \\ K_{o1}^T & K_d & K_{o1} & K_{o2} & K_{o2}^T \\ K_{o2}^T & K_{o1}^T & K_d & K_{o1} & K_{o2} \\ K_{o2} & K_{o2}^T & K_{o1}^T & K_d & K_{o1} \\ K_{o1} & K_{o2} & K_{o2}^T & K_{o1}^T & K_d \end{bmatrix} \quad (2)$$

211 As a result, when solving the eigenvalue problem of the Koopman matrix $\mathbf{K}v = \lambda v$, the **eigenvectors**
 212 **N-dimensional eigenvector v** can be decomposed as $[v_A, v_B, v_C, v_D, v_E]$. **The equation becomes $\mathbf{K}_d \cdot v_A +$**
 213 **concatenation of v_A, v_B, v_C, v_D, v_E ($N/5$ -dimensional). The first component of the equation becomes**
 214 **$\mathbf{K}_d \cdot v_A + \mathbf{K}_{o1} \cdot v_B + \mathbf{K}_{o2} \cdot v_C + \mathbf{K}_{o2}^\top \cdot v_D + \mathbf{K}_{o1}^\top \cdot v_E = \lambda v_A$.**

215 To enhance data efficiency and reduce estimation error in a large kinetic model (see the toy model
 216 system in **Figure S1**, **Figure S2**, and **Figure S3**), we aim to understand the system dynamics un-
 217 der degeneracy. This involves seeking a symmetry-adapted mapping that remains invariant un-
 218 der the symmetry group. In this case, we have $v_A = v_B = v_C = v_D = v_E$, and the eigenvalue
 219 problem can be simplified to **$\mathbf{K}^\top v_A = \lambda v_A$, where $\mathbf{K}^\top = \mathbf{K}_d + \mathbf{K}_{o1} + \mathbf{K}_{o2} + \mathbf{K}_{o2}^\top + \mathbf{K}_{o1}^\top$** , **$\mathbf{K}_{\text{sum}} v_A = \lambda v_A$, where**
 220 **$\mathbf{K}_{\text{sum}} = \mathbf{K}_d + \mathbf{K}_{o1} + \mathbf{K}_{o2} + \mathbf{K}_{o2}^\top + \mathbf{K}_{o1}^\top$.**

221 The features x_i can be projected onto the dominant eigenspace (independent components, ICs) of
 222 **\mathbf{K} by direct by summing over all five subunits (sub-ICs), subsystems (subICs):**

$$x_i = \begin{bmatrix} x_{iA} \\ x_{iB} \\ x_{iC} \\ x_{iD} \\ x_{iE} \end{bmatrix}, \quad \text{subICs} = \begin{bmatrix} v_A \cdot x_{iA} \\ v_A \cdot x_{iB} \\ v_A \cdot x_{iC} \\ v_A \cdot x_{iD} \\ v_A \cdot x_{iE} \end{bmatrix}, \quad (3)$$

223

$$\text{ICs} = v_A \cdot x_{iA} + v_A \cdot x_{iB} + v_A \cdot x_{iC} + v_A \cdot x_{iD} + v_A \cdot x_{iE} \quad (4)$$

R2.C5. Page 6, 208 “symmetry-aware MDS” I could not find any description of this method in the literature, so more details should be presented. The authors write “[...] the adjacency matrix is calculated from the minimum Euclidean distances among all permutations, allowing for the estimation of symmetry”. First of all is not clear what “Euclidean distances” means. Did the authors calculate the distance in $3N$ dimensions where N is the total number of atoms in the receptor? or rather in the feature-space used for the MSM, i.e. inter-atomic distances? Furthermore, “adjacency matrix” is usually reserved for matrices with binary entries (contact-no contact), while MDS embeds distances, so I’d use “distance matrix”. Can the authors comment on how calculating the distance as the minimum over subunit permutations helps estimate the symmetry of the system?

Thanks for the suggestion to expand on the method description of MDS. We have now rewritten the **Methods section** for the symmetry-aware MDS methodology (p. 6 l. 225), as detailed above in response to R1C8. As also described there, our example notebook on GitHub (https://github.com/yuxuanzhuang/sym_msm/blob/main/example/example_sym_mds.ipynb) demonstrates that when introducing asymmetry to one of the subsystems in a toy Gaussian system, symmetry-aware multidimensional scaling (MDS) can accurately capture the scale of the asymmetry (Reviewer Figure 2). We have also included a block of algorithmic explanation to clarify how distances between samples are calculated. It specifically computes the Euclidean distances using the array formed from the subIC values:

Algorithm 1 Get distance between two samples for five-fold symmetric systems

```

 $S_i \leftarrow (S_{iA}, S_{iB}, S_{iC}, S_{iD}, S_{iE})$ 
 $S_j \leftarrow (S_{jA}, S_{jB}, S_{jC}, S_{jD}, S_{jE})$ 
 $D \leftarrow inf$ 
for  $i \leftarrow 1$  to 5 do
     $S_j \leftarrow roll(S_j, 1)$ 
     $D \leftarrow min(D, Euclidean(S_i, S_j))$ 
end for
return  $D$ 

```

R2.C6. Figure 2 and figure 6, panel D. I think the axis label is misleading (should be “first-passage-time”): there should be only one *mean* first-passage-time. Moreover, the MSM is based on microstates, whereas the plot shows the transitions between coarse-grained states. It would be clearer if the authors clarified in detail how the histogram was “estimated from the MSM”. Are the results in the histogram (panel D) in agreement with panel C? For instance, in fig 2C, O->D is 1.8 ms but there is no density in panel D beyond 1.6 ms.

Thank you for highlighting this issue. We should clarify that the D panels in Figures 2 and 6 present error estimations for the mean first passage times (MFPT) derived from the Bayesian MSM, not the first-passage time itself. The discrepancy between panel D and panel C arose from a mix-up of results obtained using pyEMMA in panel D and deeptime in panel C in the course of multimodal validation. Although small differences between the two software packages are noticeable for slow transitions (as shown in Reviewer Figure 3), and of unclear origin. To eliminate any confusion, we have updated the results in the D panels to those derived from deeptime, and adjusted the legend accordingly.

Reviewer Figure 3. State transition times calculated using deeptime (left) or pyEMMA (right) MSM tools. Above, mean first-passage times (MFPT, arrows) between macrostates (circles), with circle size indicating its relative stability. Passage times for transitions beyond the classical gating cycle (C-F-O-D-I-C) are hidden for clarity. Below, MFPT histograms showing transitions between states O and F (blue) and between states O and D (orange), estimated from Bayesian MSMs (100 samples).

Erik Lindahl
Professor of Biophysics

Stockholm, 5 May 2024

R2.C7. Page 8, line 295 “when we projected existing 7 cryo-EM structures onto the IC space, one particular structure resolved in the presence of EVP-6124 and PNU fell into the basin corresponding to the F state” It would be interesting to see where the other structures (3?) end up

Thank you for your suggestion. We realized that there was a typo: there are actually only 6 really different cryo-EM structures in total. The other two structures, 7EKI (apo) and 7EKP (EVP), share sub-Å RMSD values with the projected structures 7KOO (BGT, closed) and 7KOQ (EPJ, desensitized). We omitted the projection of these two structures for clarity. We have modified the manuscript in light of this:

p. 9 l. 317: **Interestingly, an additional cryo-EM structure obtained in the presence of EVP-6124 and PNU, with a distinctive conformation compared to those used for initial pathway generation, fell into the basin corresponding to the F state.**

Reviewer #2 (Remarks on code availability):

R2.C8. I commend the authors for sharing their code. I am well aware of the amount of work it takes to take a functioning code and make it available and usable by other researchers.

I have browsed the code in the repositories, and could install the `sym_msm` library via pip.

However, the documentation for both repositories is still missing navigating the code without comments would take more effort that I can devote to this review, so I cannot comment on the reproducibility of the results.

We fully agree that providing reproducible code alongside a manuscript is crucial. Therefore, we have highlighted these repositories in a separate “Code availability” section, and expanded the README documentation for both repositories to help future users get started. The `sym_msm` library contains generic code applicable to other symmetric systems, and we have included toy models in the *example* section. These can be used to run tests on synthetic code and to perform comparisons with regular TICA. The `msm_a7_nachrs` library, on the other hand, contains more system-specific code to build the system, run equilibrations, and extract features from trajectories. To reproduce the results presented in our manuscript, all relevant notebooks, and package archives can be found on Zenodo at <https://zenodo.org/records/11117001>.

Thank you once more for the highly constructive and detailed feedback from both reviewers; they have significantly improved the quality of the work - and we apologize for the few typos and mistakes they noticed.

Erik Lindahl

REVIEWERS' COMMENTS

Reviewer #1 (Remarks to the Author):

All my revision requests have been incorporated. I think the manuscript is now ready for publication in Nature Communications.

Reviewer #1 (Remarks on code availability):

I appreciate the publication of the analysis code. I did not review the code itself, as I think this is beyond the time I can attribute to a paper review, but think the open source community is a sufficient basis for further reviewing the code when it will be applied by others.